



# Marine terraces of the last interglacial period along the Pacific coast of South America (1°N-40°S)

Roland Freisleben[1], Julius Jara-Muñoz[1], Daniel Melnick[2,3], José Miguel Martínez[2,3], Manfred R. Strecker[1]

[1] Institut für Geowissenschaften, Universität Potsdam, 14476 Potsdam, Germany
[2] Instituto de Ciencias de la Tierra, TAQUACH, Universidad Austral de Chile, Valdivia, Chile
[3] Millennium Nucleus The Seismic Cycle Along Subduction Zones, Valdivia, Chile

*Correspondence to*: Roland Freisleben (freisleb@uni-potsdam.de)

**Abstract.** Tectonically active coasts are dynamic environments characterized by the presence of multiple marine terraces formed by the combined effects of wave-erosion, tectonic uplift, and sea-level oscillations at glacial-cycle timescales. Well-preserved erosional terraces from the last interglacial sea-level highstand are ideal marker horizons for reconstructing past sea-level positions and calculating vertical displacement rates. We carried out an almost continuous mapping of the last interglacial marine terrace along ~5,000 km of the western coast of South America between 1°N and 40°S. We used quantitatively replicable approaches constrained by published terrace-age estimates to ultimately compare elevations and patterns of uplifted terraces with tectonic and climatic parameters in order to evaluate the controlling mechanisms for the formation and preservation of marine terraces, and crustal deformation. Uncertainties were estimated on the basis of measurement errors and the distance from referencing points. Overall, our results indicate a median elevation of 30.1 m, which would imply a median uplift rate of 0.22 m/ka averaged over the past ~125 ka. The patterns of terrace elevation and uplift rate display high-amplitude (~100–200 m) and long-wavelength (~$10^2$ km) structures at the Manta Peninsula (Ecuador), the San Juan de Marcona area (central Peru), and the Arauco Peninsula (south-central Chile). Medium-wavelength structures occur at the Mejillones Peninsula and Topocalma in Chile, while short-wavelength (< 10 km) features are for instance located near Los Vilos, Valparaíso, and Carranza, Chile. We interpret the long-wavelength deformation to be controlled by deep-seated processes at the plate interface such as the subduction of major bathymetric anomalies like the Nazca and Carnegie ridges. In contrast, short-wavelength deformation may be primarily controlled by sources in the upper plate such as crustal faulting, which, however, may also be associated with the subduction of topographically less pronounced bathymetric anomalies. Latitudinal differences in climate additionally control the formation and preservation of marine terraces. Based on our synopsis we propose that increasing wave height and tidal range result in enhanced erosion and morphologically well-defined marine terraces in south-central Chile. Our study emphasizes the importance of using systematic measurements and uniform, quantitative methodologies to characterize and correctly interpret marine terraces at regional scales, especially if they are used to unravel tectonic and climatic forcing mechanisms of their formation. This database is an integral part of the World Atlas of Last Interglacial Shorelines (WALIS), published online at http://doi.org/10.5281/zenodo.4309748 (Freisleben et al., 2020).



## 1. Introduction

Tectonically active coasts are highly dynamic geomorphic environments and they host densely-populated centers and
associated infrastructure (Melet et al., 2020). Coastal areas have been episodically affected by the effects of sea-level
changes at glacial timescales, drastically modifying the landscape and leaving behind fossil geomorphic markers, such
as former paleo-shorelines, abrasion platforms, and marine terraces (Lajoie, 1986). One of the most prominent coastal
landforms are marine terraces that were generated during the protracted last interglacial sea-level highstand occurred
~125 ka ago (Siddall et al., 2006). These terraces are characterized by a higher preservation potential, which facilitates
their recognition, mapping and lateral correlation. Furthermore, because of their high degree of preservation and
relatively young age, they have been used to estimate vertical deformation rates at local and regional scales. The
relative abundance and geomorphic characteristics of the last interglacial marine terraces make them ideal geomorphic
markers with which to reconstruct past sea-level positions and to enable comparisons between distant sites under
different climatic and tectonic settings.
The Western South American Coast (WSC) is a tectonically active region that has been repeatedly affected by
megathrust earthquakes and associated surface deformation (Beck et al., 1998; Melnick et al., 2006; Bilek, 2010;
Baker et al., 2013). Interestingly, previous studies have shown that despite the broad spectrum of latitudinal climatic
conditions and erosional regimes along the WSC, marine terraces are scattered, but omnipresent along the coast (Ota
et al., 1995; Rehak et al., 2010; Bernhardt et al., 2016; Melnick, 2016; Bernhardt et al., 2017). However, only a few
studies on interglacial marine terraces have been conducted along the WSC, primarily in specific areas where they are
best expressed; this has resulted in disparate and inconclusive marine terrace measurements based on different
methodological approaches and ambiguous interpretations concerning their origin in a tectonic and climatic context
(Hsu et al., 1989; Ortlieb and Macharé, 1990; Hsu, 1992; Macharé and Ortlieb, 1992; Pedoja et al., 2006b; Saillard et
al., 2009; Regard et al., 2010; Saillard et al., 2011; Rodríguez et al., 2013; Pedoja et al., 2014). This lack of reliable
data points has revealed a need to re-examine the last interglacial marine terraces along the WSC based on standardized
methodologies in order to obtain a systematic and continuous record of marine terrace elevations along the coast. This
information is crucial in order to increase our knowledge of the climatic and tectonic forcing mechanisms that
contributed to the formation and degradation of marine terraces in this region.
Marine terrace sequences at tectonically active coasts are landforms formed by wave erosion and/or accumulation of
sediments resulting from the interaction between tectonic uplift and superposed oscillating sea-level changes (Lajoie,
1986; Anderson et al., 1999; Jara-Muñoz et al., 2015). Typically, marine terrace elevations are estimated based on the
shoreline angle. The marine terrace morphology comprises a gently inclined marine abrasion platform or depositional
surface that terminates landward at a steeply sloping paleo-cliff surface. The intersection point between both surfaces
represents the sea-level position during the formation of the marine terrace also known as shoreline angle; if coastal
uplift is rapid, such uplifting abrasion or depositional surfaces may be preserved in the landscape and remain unaltered
by the effects of subsequent sea-level oscillations (Lajoie, 1986).





The analysis of elevation patterns based on shoreline-angle measurements at subduction margins has been largely used
to estimate vertical deformation rates and the mechanisms controlling deformation, including the interaction of the
upper plate with bathymetric anomalies, the activity of crustal faults in the upper plate, and deep-seated processes
such as basal accretion of subducted trench sediments (Taylor et al., 1987; Hsu, 1992; Macharé and Ortlieb, 1992; Ota
et al., 1995; Saillard et al., 2011; Pedoja et al., 2014; Jara-Muñoz et al., 2015; Melnick, 2016). The shoreline angle
represents a 1D descriptor of the marine terrace elevation, whose measurements are reproducible when using
quantitative morphometric approaches (Jara-Muñoz et al., 2016). Furthermore, the estimation of the marine terrace
elevations based on shoreline angles can be further improved by quantifying their relationship with paleo-sea level,
also known as the indicative meaning (Lorscheid and Rovere, 2019).
In this continental-scale compilation of marine terrace elevations along the WSC, we present systematically mapped
shoreline angles of marine terraces of the last (Eem/Sangamon) interglacial obtained along 5,000 km of coastline
between 1°N and 40°S. In this synthesis we rely on chronological constraints from previous regional studies and
compilations (Pedoja et al., 2014). For the first time we are able to introduce an almost continuous pattern of terrace
elevation and coastal uplift rates at a spatial scale of $10^3$ km along the WSC. Furthermore, in our database we compare
tectonic and climatic parameters to elucidate the mechanisms controlling the formation and preservation of marine
terraces, and patterns of crustal deformation along the coast. This study was thus primarily intended to provide a
comprehensive, standardized database and description of last interglacial marine terrace elevations along the
tectonically active coast of South America. This database therefore affords future research into coastal environments
to decipher potential tectonic forcings with regard to the deformation and seismotectonic segmentation of the forearc;
as such this database will ultimately help to decipher the relationship between upper-plate deformation, vertical motion
and bathymetric anomalies and aid in the identification of regional fault motions along pre-existing anisotropies in the
South American continental plate. Finally, our database includes information on climate-driving forcing mechanisms
that may influence the formation, modification and/or destruction of marine terraces in different climatic sectors along
the South American convergent margin. This new database is part of the World Atlas of Last Interglacial Shorelines
(WALIS), published online at http://doi.org/10.5281/zenodo.4309748 (Freisleben et al., 2020).
**2.    Geologic and geomorphic setting of the WSC**
**2.1.    Tectonic and seismotectonic setting**
**2.1.1.    Subduction geometry and bathymetric features**
The tectonic setting of the convergent margin of South America is controlled by subduction of the oceanic Nazca plate
beneath the South American continental plate. The convergence rate varies between 66 mm/a in the north (8°S latitude)
and 74 mm/a in the south (27°S latitude) (Fig. 1). The convergence azimuth changes slightly from N81.7° toward
N77.5° from north to south (DeMets et al., 2010). The South American subduction zone is divided into four major
segments inferred from the spatial distribution of Benioff seismicity (Barazangi and Isacks, 1976; Jordan et al., 1983)
(Fig. 1). The segments beneath northern and central Peru (2°–15°S) and beneath central Chile (27°–33°S) are



characterized by a gentle dip of the subducting plate between 5° and 10° at depths of ~100 km (Hayes et al., 2018),
whereas the segments beneath southern Peru and northern Chile (15°–27°S), and beneath southern Chile (33°–45°S)
have steeper dips of 25° to 30°. Spatial distributions of earthquakes furthermore indicate a steep-slab subduction
segment in Ecuador and southern Colombia (2°S to 5°N), and a flat-slab segment in NW Colombia (north of 5°N)
(Pilger, 1981; Cahill and Isacks, 1992; Gutscher et al., 2000; Ramos and Folguera, 2009). Processes that have been
inferred to be responsible for the shallowing of the subduction slab include the subduction of large buoyant ridges or
plateaus (Espurt et al., 2008) as well as the combination of trenchward motion of thick, buoyant cratonic lithosphere
accompanied by trench retreat (Sobolev and Babeyko, 2005; Manea et al., 2012). Volcanic activity as well as the
forearc architecture and distribution of upper-plate deformation further emphasize the location of flat-slab subduction
segments (Jordan et al., 1983; Kay et al., 1987; Ramos and Folguera, 2009).
Several bathymetric anomalies have been recognized on the subducting Nazca plate. The two most prominent
anomalies being subducted beneath South America are the Carnegie and Nazca aseismic ridges at 0° and 15°S,
respectively. The Carnegie Ridge subducts roughly parallel with the convergence direction and its position should
have remained relatively stable beneath the continent (Angermann et al., 1999; Gutscher et al., 1999; DeMets et al.,
2010; Martinod et al., 2016a). In contrast, the obliquity of the Nazca Ridge with respect to the convergence direction
resulted in 500 km SE-directed migration of the subduction locus during the last 10 Ma (Hampel, 2002; Saillard et al.,
2011; Martinod et al., 2016a). Similarly, smaller aseismic ridges such as the Juan Fernández Ridge and the Iquique
Ridge subduct beneath the South American continent at 32°S and 21°S, respectively. The subduction of these
bathymetric anomalies are thought to influence the characteristics of interplate coupling and seismic rupture (Bilek et
al., 2003; Wang and Bilek, 2011; Geersen et al., 2015; Collot et al., 2017) and mark the boundaries between flat and
steep subduction segments and changes between subduction erosion and accretion (Jordan et al., 1983; von Huene et
al., 1997; Ramos and Folguera, 2009) (Fig. 1).
In addition to bathymetric anomalies, several studies have shown that variations in the amount of sediments in the
trench may control the subduction regime from an erosional mode to an accretionary mode (von Huene and Scholl,
1991; Bangs and Cande, 1997). In addition, the amount of sediment in the trench has also been hypothesized to
influence the style of interplate seismicity (Lamb and Davis, 2003). At the southern Chile margin, thick trench
sediments and a steeper subduction angle correlate primarily with subduction accretion, although the area of the
intercept of the continental plate with the Chile Rise spreading center locally exhibits the opposite case (von Huene
and Scholl, 1991; Bangs and Cande, 1997). Subduction erosion characterizes the region north of the southern volcanic
zone from central and northern Chile to southern Peru (33°–15°S) due to decreasing sediment supply to the trench,
especially within the flat-slab subduction segments (Stern, 1991; von Huene and Scholl, 1991; Bangs and Cande,
1997; Clift and Vannucchi, 2004). Clift and Hartley (2007) and Lohrmann et al. (2003) argued for an alternate style
of slow tectonic erosion leading to underplating of subducted material below the base of the crustal forearc,
synchronous with tectonic erosion beneath the trenchward part of the forearc. For the northern Andes, several authors
also classify the subduction zone as an erosional type (Clift and Vannucchi, 2004; Scholl and Huene, 2007; Marcaillou
et al., 2016).

### 2.1.2. Major continental fault systems in the coastal realm

The South American convergent margin comprises several fault systems with different kinematics. Here we summarize the main structures that affect the Pacific coastal areas. North of the Talara bend (5°S), active thrusting and dextral strike-slip faulting dominates. The most prominent dextral fault in this region is the 2000-km-long, northeast-striking Dolores-Guayaquil megashear (DGM), which starts in the Gulf of Guayaquil and terminates in the Colombian hinterland east of the range-bounding thrust faults of the Colombian Andes (Veloza et al., 2012; Villegas-Lanza et al., 2016) (Fig. 1). The coastal lowlands of Ecuador comprise several thrust faults, while normal faulting in the Gulf of Guayaquil and dextral strike-slip faulting at the Santa Elena Peninsula can be observed farther south (Veloza et al., 2012). Normal faults have been described along the coast of Peru at the Illescas Peninsula in the north (6°S), in the San Juan de Marcona area with the El Huevo–Lomas fault system (14.5°–16°S), and the Incapuquio fault system in the south (17°–18°S) (Veloza et al., 2012; Villegas-Lanza et al., 2016). The main fault zones of the Chilean convergent margin comprise the Atacama Fault System (AFS) in the Coastal Cordillera extending from Iquique to La Serena (29.75°S) with predominantly N-S-striking normal faults, which result in relative uplift of their western side (e.g., Mejillones fault, Salar del Carmen fault) (Naranjo, 1987; González and Carrizo, 2003; Cembrano et al., 2007). Smaller coastal fault systems farther south are located in the Altos de Talinay area (30.5°S, Puerto Aldea fault), near Valparaíso (33°S, Quintay and Valparaíso faults), near the Arauco Peninsula (36°–39°S, Santa María and Lanalhue faults), and in between (Topocalma, Pichilemu, Carranza, and Pelluhue faults) (Ota et al., 1995; Melnick et al., 2009; Santibáñez et al., 2019; Maldonado et al., 2020; Melnick et al., 2020). However, there is still limited knowledge regarding Quaternary slip rates and kinematics and, most importantly, the location of active faults along the forearc region of South America (Jara-Muñoz et al., 2018; Melnick et al., 2019).

## 2.2. Climate and geomorphic setting

### 2.2.1. Geomorphology

The 8000-km-long Andean orogen is a major, hemisphere-scale feature that can be divided into different segments with distinctive geomorphic and tectonic characteristics. The principal segments comprise the NNE-SSW trending Colombian-Ecuadorian segment (12°N–5°S), the NW-SE oriented Peruvian segment (5°–18°S), and the N-S trending Chilean segment (18°–56°S) (Jaillard et al., 2000) (Fig. 1). Two major breaks separate these segments; these are the Huancabamba bend in northern Peru and the Arica bend at the Peru-Chile border. The distance of the trench from the WSC coastline averages 118 km and ranges between 44 and 217 km. The depth of the trench fluctuates between 2920 and 8177 m (GEBCO Bathymetric Compilation Group, 2020), and the continental shelf has an average width of 28 km (Paris et al., 2016).

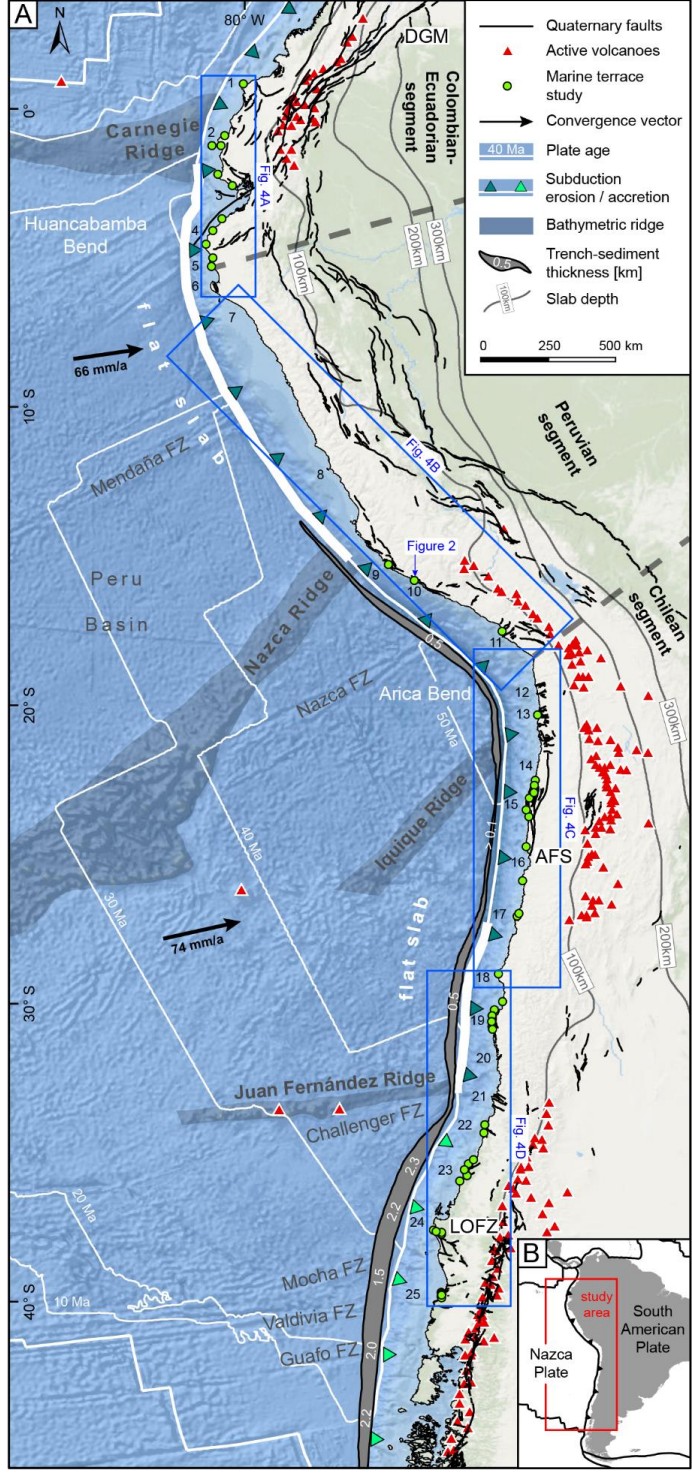

Figure 1. (A) Morphotectonic setting of the South American margin showing major fault systems/crustal faults (Costa et al., 2000; Veloza et al., 2012; Melnick et al., 2020), slab depth (Hayes et al., 2018), and flat-slab subduction segments, active volcanos (Venzke, 2013), bathymetric features of the subducting plate, trench-sediment thickness (Bangs and Cande, 1997), segments of subduction erosion and accretion (Clift and Vannucchi, 2004), plate age (Müller et al., 2008), convergence vectors (DeMets et al., 2010), and marine terrace ages used for lateral correlation. DGM: Dolores-Guayaquil megashear, AFS: Atacama Fault System, LOFZ: Liquiñe-Ofqui Fault Zone, FZ: Fracture Zone, 1: Punta Galera, 2: Manta Pen., 3: Santa Elena Pen., 4: Tablazo Lobitos, 5: Paita Pen., 6: Illescas Pen., 7: Chiclayo, 8: Lima, 9: San Juan de Marcona, 10: Chala Bay, 11: Pampa del Palo, 12: Pisagua, 13: Iquique, 14: Tocopilla, 15: Mejillones Pen., 16: Taltal, 17: Caldera, 18: Punta Choros, 19: Altos de Talinay, 20: Los Vilos, 21: Valparaíso, 22: Topocalma, 23: Carranza, 24: Arauco Pen., 25: Valdivia (World Ocean Basemap: Esri, Garmin, GEBCO, NOAA NGDC, and other contributors). (B) Location of the study area.






In the 50- to 180-km-wide coastal area of the Ecuadorian Andes, where the Western Cordillera is flanked by a
structural depression, relief is relatively low (< 300 m asl). The Gulf of Guayaquil (3°S) and the Dolores-Guayaquil
megashear separate the northern from the southern forearc units. The coast-trench distance along the Huancabamba
bend is quite small (~55–90 km), except for the Gulf of Guayaquil, and the trench east of the Carnegie Ridge is at a
relatively shallow depth of ~3.5 km. Farther south, the Peruvian forearc comprises the up to 160-km-wide Coastal
Plains in the north and the narrow, 3000-m-high Western Cordillera. While the Coastal Plains in north-central Peru
are relatively narrow (< 40 km), they widen in southern Peru, and the Western Cordillera increases to more than 5000
m (Suárez et al., 1983; Jaillard et al., 2000). The region between the coast and the trench in central Peru (up to 220
km) narrows toward the San Juan de Marcona area (~75 km) near the Nazca Ridge, and the relatively deep trench
(~6.5 km) becomes shallower (< 5 km) (GEBCO Bathymetric Compilation Group, 2020). Between 18°S and 28°S,
the Chilean forearc comprises the 50-km-wide and up to 2700-m-high Coastal Cordillera, which is separated from the
Precordillera by the Central Depression. In the flat-slab subduction segment between 27°S and 33°S there is neither a
morphotectonic region characterized by a central depression nor active volcanism in the high Andean cordillera (Fig.
1) (Jordan et al., 1983). The Chilean forearc comprises the Coastal Cordillera, which varies in altitude from up to 2000
m at 33°S to 500 m at 46°S, and the Central Depression that separates the forearc from the Main Cordillera. From the
Arica bend, where the coast-trench distance is up to 170 km and the trench ~8 km deep, a slight increase in distance
can be observed in Chile toward the south (~80–130 km), as can a decrease in trench depth to ~4.5 km.
**2.2.2.    Marine terraces and coastal uplift rates**
Wave erosion forms wave-cut terrace levels, while the accumulation of shallow marine sediments during sea-level
highstands forms wave-built terraces. Another type of terrace is known as "rasa" and refers to wide shore platforms
formed under slow-uplift conditions (< 0.2 m/ka), and the repeated reoccupation of this surface by high sea levels
(Anderson et al., 1999; Regard et al., 2010; Rodríguez et al., 2013; Melnick, 2016). Typically, the formation of
Pleistocene marine terraces in the study area occurred during interglacial and interstadial relative sea-level highstands
that were superposed on the uplifting coastal areas; according to the Quaternary oxygen-isotope curve defining warm
and cold periods, high Quaternary sea levels have been correlated with warm periods and are denoted with the odd-
numbered Marine Isotope Stages (MIS) (Lajoie, 1986; Shackleton et al., 2003).
Along the WSC, staircase-like sequences of multiple marine terraces are preserved nearly continuously along the
coast. They comprise primarily wave-cut surfaces that are frequently covered by beach ridges of siliciclastic sediments
and local accumulations of carbonate bioclastic materials along the beach ridges (Ota et al., 1995; Saillard et al., 2009;
Rodríguez et al., 2013; Martinod et al., 2016b). Rasa surfaces exist in the regions of southern Peru and northern Chile
(Regard et al., 2010; Rodríguez et al., 2013; Melnick, 2016). Particularly the well-preserved MIS-5e terrace level has
been largely used as a strain marker in the correlation of uplifted coastal sectors due to its lateral continuity and high
potential for preservation. Global observations of sea-level fluctuations during MIS-5 differentiate between three
second-order highstands at 80 ka (5a), 105 ka (5c), and 128 to 116 ka (5e) with paleo-sea levels of -20 m for both of





the younger and +3 ± 3 m for the oldest highstand (Stirling et al., 1998; Siddall et al., 2006; Hearty et al., 2007;
Rohling et al., 2009; Pedoja et al., 2014). The database generated in this study is based exclusively in the last
interglacial marine terraces exposed along the WSC, between Ecuador and Southern Chile (1°S to 40°S). In the
following section we present a brief review of previously studied marine terrace sites in this area.
Paleo-shoreline elevations of the last interglacial (MIS-5e) in Ecuador are found at elevations of around 45 ± 2 m asl
in Punta Galera (Esmeraldas area), 43–57 ± 2 m on the Manta Peninsula and La Plata Island, and 15 ± 5 m asl on the
Santa Elena Peninsula (Pedoja et al., 2006b; Pedoja et al., 2006a). In northern Peru, MIS-5e terraces have been
described at elevations of 18–31 m asl for the Tablazo Lobitos (Cancas and Mancora areas), at 25 ± 5 m asl on the
Paita Peninsula, and at 18 ± 3 m asl on the Illescas Peninsula and the Bay of Bayovar (Pedoja et al., 2006b). Farther
south, MIS-5e terraces are exceptionally high in the San Juan de Marcona area immediately south of the subducting
Nazca Ridge, with maximum elevations of 80 m at the Cerro Tres Hermanas and 105 m at the Cerro El Huevo (Hsu
et al., 1989; Ortlieb and Macharé, 1990; Saillard et al., 2011). The Pampa del Palo region in southern Peru shows
relatively thick vertical stacks of shallow marine terrace deposits related to MIS-7, 5e (~20 m), and 5c that may
indicate a different geodynamic behavior compared to adjacent regions (Ortlieb et al., 1996). In central and northern
Chile, the terrace levels of the last interglacial exist at 250–400 m, 150–240 m, 80–130 m, and 30–40 m, and in
southern Chile at 170–200 m, 70 m, 20–38 m, 8–10 m (Fuenzalida et al., 1965). Specifically, between 24°S and 32°S,
paleo-shoreline elevations of the last interglacial (MIS-5e) range between 25 and 45 m (Ota et al., 1995; Saillard et
al., 2009; Martinod et al., 2016b). Shore platforms are higher in the Altos de Talinay area (30.3°–31.3°S) but are small,
poorly preserved, and terminate at a high coastal scarp between 26.75°S and 24°S (Martinod et al., 2016b). Shoreline-
angle elevations within the Maule segment (34°–38°S) vary from high altitudes in the Arauco and Topocalma areas
(200 m) to moderate elevations near Caranza (110 m) and very low elevations in between (15 m) (Melnick et al., 2009;
Jara-Muñoz et al., 2015).
Coastal uplift-rate estimates along the WSC mainly comprise calculations for the Talara Arc, the San Juan de Marcona
area, the Mejillones Peninsula, the Altos de Talinay area, and several regions in south-central Chile. Along the Talara
Arc (6.5°S to 1°N), marine terraces of the Manta Peninsula and La Plata Island in central Ecuador indicate the most
extensive uplift rate of 0.31 to 0.42 m/ka, while lower uplift rates are documented to the north in the Esmeraldas area
(0.34 m/ka) and especially to the south at the Santa Elena Peninsula (0.1 m/ka). In northern Peru, uplift rates are
relative low, ranging from 0.17–0.21 m/ka for the Tablazo Lobitos and 0.16 m/ka for the Paita Peninsula, to 0.12 m/ka
for the Bay of Bayovar and the Illescas Peninsula (Pedoja et al., 2006b; Pedoja et al., 2006a). Marine terraces above
the subducting Nazca Ridge (13.5°–15.6°S) show variations in uplift rate where the coastal forearc above the northern
flank of the ridge is stable or has undergone net subsidence, or where the coast above the ridge crest is rising at about
0.3 m/ka and the coast above the southern flank (San Juan de Marcona) is uplifting at a rate of 0.5 m/ka (Hsu, 1992)
or at even 0.7 m/ka for at least the last 125 ka according to Ortlieb and Macharé (1990). Saillard et al. (2011) state
that long-term regional uplift in the San Juan de Marcona area has increased since about 800 ka related to the
southward migration of the Nazca Ridge, and ranges from 0.44 to 0.87 m/ka. The Pampa del Palo area in southern
Peru rose more slowly or was even down-faulted and had subsided with respect to the adjacent coastal regions (Ortlieb





et al., 1996). These movements ceased after the highstand during the MIS-5e and slow uplift rates of around 0.16 m/ka
characterized the region since 100 ka (Ortlieb et al., 1996). In northern Chile (24°–32°S), uplift rates for the Late
Pleistocene average around 0.28 ± 0.15 m/ka (Martinod et al., 2016b), except for the Altos de Talinay area, where
pulses of rapid uplift occurred during the Middle Pleistocene (Ota et al., 1995; Saillard et al., 2009; Martinod et al.,
2016b). The Central Andean rasa (15°–33°S) and the oldest Pleistocene shore platforms – which are also generally
wider – indicate accelerated and spatially continuous uplift after a period of tectonic stability or subsidence. According
to Melnick (2016), the Central Andean rasa has experienced slow and steady long-term uplift at 0.13 ± 0.04 m/ka
during the Quaternary, predominantly accumulating strain through deep earthquakes at the crust-mantle boundary
(Moho) below the locked portion of the plate interface. The lowest uplift rates occur at the Arica bend and increase
gradually southward; the highest values are reached along geomorphically distinct peninsulas (Melnick, 2016). In the
Maule segment, the mean uplift rate for the MIS-5 terrace level is 0.5 m/ka, exceeded only in the areas of Topocalma,
Carranza, and Arauco (reaching up to 1.6 m/ka) (Melnick et al., 2009; Jara-Muñoz et al., 2015). Although there are
several studies of marine terraces along the WSC, these are isolated and based on different methodological approaches,
mapping and leveling resolution, as well as dating techniques, which makes regional comparisons and correlations
difficult.

### 254 2.2.3. Climate

Apart from latitudinal temperature changes, the present-day morphotectonic provinces along the South American
margin have a pronounced impact on the precipitation gradients on the west coast of South America. Since mountain
ranges are oriented approximately perpendicular to moisture-bearing winds, they affect both flanks of the orogen
(Strecker et al., 2007). The regional-scale pattern of wind circulation is dominated by westerly winds at
subtropical/extratropical latitudes primarily up to about 27°S (Garreaud, 2009). However, anticyclones over the South
Pacific result in winds blowing from the south along the coast between 35°S and 10°S (Garreaud, 2009). The moisture
in the equatorial Andes (Ecuador and Colombia) and in the areas farther south (27°S) is fed by winds from the Amazon
basin and the Gulf of Panama, resulting in rainfall mainly on the eastern flanks of the mountain range (Bendix et al.,
2006; Bookhagen and Strecker, 2008; Garreaud, 2009). The Andes of southern Ecuador, Peru, and northern Chile are
dominated by a rain-shadow effect that causes aridity within the Andean Plateau (Altiplano-Puna), the Western
Cordillera, and the coastal region (Houston and Hartley, 2003; Strecker et al., 2007; Garreaud, 2009). Furthermore,
the aridity is exacerbated by the effects of the cold Humboldt current, which prevents humidity from the Pacific from
penetrating inland (Houston and Hartley, 2003; Garreaud, 2009; Coudurier-Curveur et al., 2015). The precipitation
gradient reverses between 27°S and 35°S, where the Southern Hemisphere Westerlies cause abundant rainfall on the
western flanks of the Coastal and Main cordilleras (Strecker et al., 2007; Garreaud, 2009). Martinod et al. (2016b) has
proposed that latitudinal differences in climate largely influence coastal morphology, specifically the formation of
high coastal scarps that prevent the development of extensive marine terrace sequences. However, the details of this
relationship have not been conclusively studied along the full extent of the Pacific coast of South America.



## 3. Methods

We combined – and describe in detail below – bibliographic information, different topographic data sets, and uniform morphometric and statistical approaches to assess the elevation of marine terraces and accompanying vertical deformation rates along the South American margin.

### 3.1. Mapping marine terraces

Marine terraces are primarily described based on their elevation, which is essential for determining vertical deformation rates. The measurements of the marine terrace elevations of the last interglacial were performed using TanDEM-X topography (12 and 30 m horizontal resolution) (German Aerospace Center (DLR), 2018), and digital terrain models from LiDAR (1, 2.5, and 5 m horizontal resolution). The DEMs were converted to orthometric heights using the ellipsoid projection of the World Geodetic System (WGS1984) and the EGM2008 ($E_{EGM08}$) geoid. The orthometrically corrected DEMs were projected in Universal Transverse Mercator (UTM) projections of varying zones, namely WGS84 UTM Zone 19S for Chile, Zone 18S for southern/central Peru, and Zone 17S for northern Peru/Ecuador.

To trace the MIS-5 shoreline, we mapped its inner edge along the west coast of South America based on the TanDEM-X topography (Fig. 2A). To facilitate mapping, we used slope and hillshade maps. We correlated the inner edge mapping with the marine terraces catalog of Pedoja et al. (2014) and the references therein (section 2.2.2, Table 1). Further references used to validate MIS-5e terrace heights include Victor et al. (2011) for the Pampa de Mejillones, Martinod et al. (2016b) for northern Chile, and Jara-Muñoz et al. (2015) for the area between 34° and 38°S. The referencing point with the nearest distance to the location of our measurements served as an orientation for the MIS-5 terrace elevation in the respective area.

A rigorous assessment of marine terrace elevations is crucial for determining accurate vertical deformation rates. Since fluvial degradation and hillslope processes after the abandonment of marine terraces may alter their morphology (Anderson et al., 1999; Jara-Muñoz et al., 2015), direct measurements of terrace elevations at the inner edge (foot of the paleo-cliff) may result in overestimation of the terrace elevations and vertical deformation rates (Jara-Muñoz et al., 2015). To precisely measure the shoreline-angle elevations of the MIS-5 terrace level, we used a profile-based approach in TerraceM, a graphical user interface in MATLAB® (Jara-Muñoz et al., 2016), available at www.terrace.com. We placed swath profiles of variable width perpendicular to the previously mapped inner edge, which were used by the TerraceM algorithm to extract maximum elevations to avoid fluvial incision (Fig. 2A). North of Caleta Chañaral (29°S), we used swath profiles of 200 m width, although we occasionally used 100-m-wide profiles for narrow terrace remnants. South of 29°S, we used swath widths of 130 and 70 m. The width was chosen based on fluvial drainage densities that are associated with climate gradients. Sensitivity tests comparing shoreline-angle measurements from different swath widths in the Chala Bay show only minimal deviations of less than 0.5 m (Fig. 2C). The sections of these profiles, which represent the undisturbed paleo-platform and paleo-cliff, were picked manually and fitted by linear regression. The extrapolated intersection between both regression lines ultimately



determines the buried shoreline-angle elevation and associated uncertainty, which is derived from the 95% confidence
interval ($2\sigma$) of both regressions (Fig. 2B). In total, we measured 1843 and 110 shoreline-angle elevations of the MIS-
5e and MIS-5c terrace levels, respectively. To quantify the paleo-position of the relative sea-level elevation and the
involved uncertainty for the WALIS template, we calculated the indicative meaning using the IMCalc software from
Lorscheid and Rovere (2019).

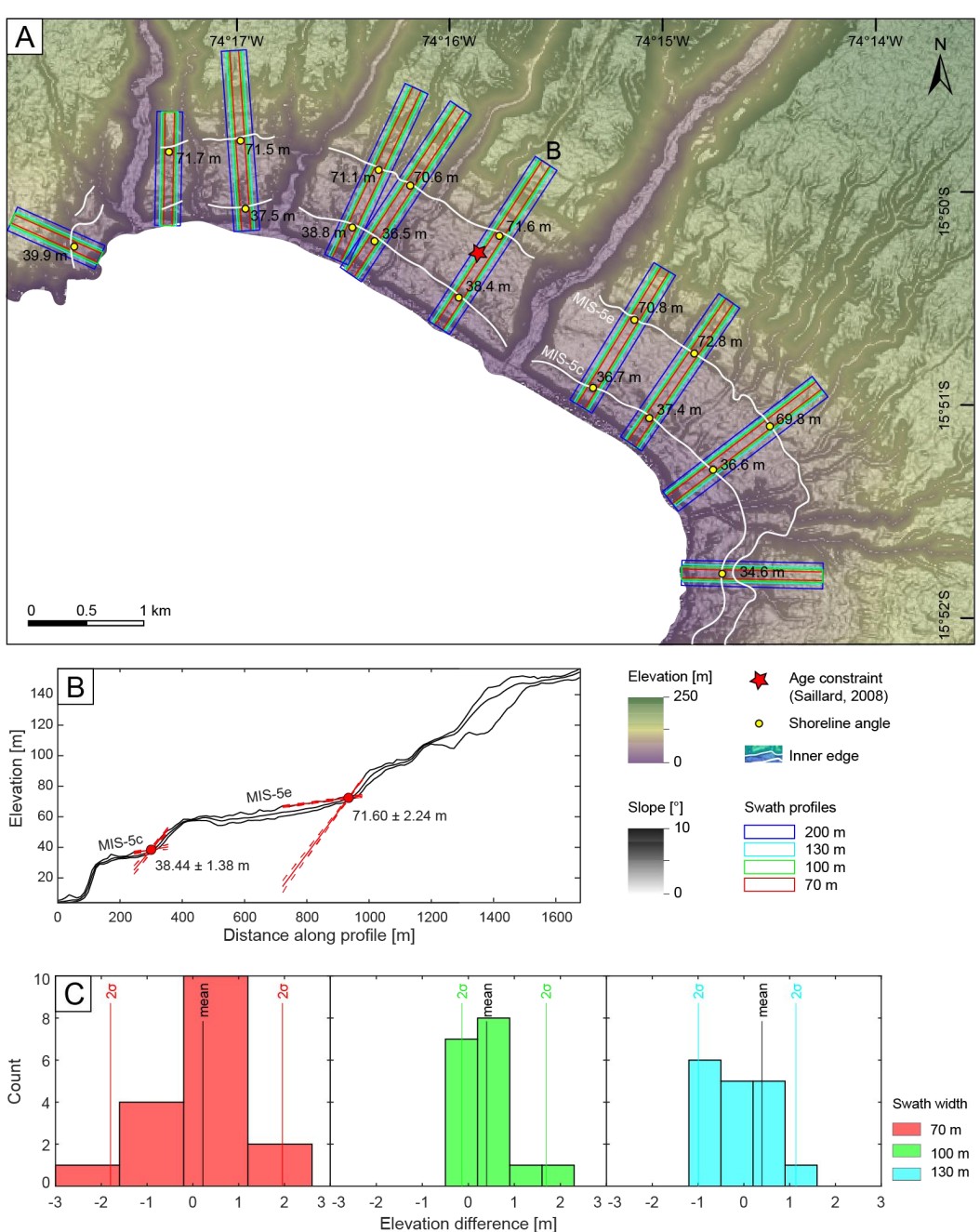

**Figure 2. (A)** Orthometrically corrected TanDEM-X and slope map of Chala Bay in south-central Peru with mapped shoreline inner edges of the MIS-5e and 5c terrace levels. Colored rectangles represent swath-profile boxes of various widths that were placed perpendicular to the inner edges for the subsequent estimation of terrace elevation in TerraceM. The red star indicates the age constraint for this area (Saillard, 2008). **(B)** Estimation of the shoreline-angle elevation in TerraceM by intersecting linear-regression fits of the paleo-cliff and paleo-platform. **(C)** Histograms of elevation differences measured for various swath widths (70 m, 100 m, and 130 m) with respect to the 200-m-wide reference swath profile (blue).





Table 1. Age constraints used for mapping of the inner edge of MIS-5 and for verifying our terrace-elevation measurements. This compilation is mainly based on the terrace catalog of Pedoja et al. (2014); added references include Victor et al. (2011) for Pampa de Mejillones, Martinod et al. (2016b) for northern Chile, and Jara-Muñoz et al. (2015) for south-central Chile. IRSL: Infrared Stimulated Luminescence, AAR: Amino-Acid Racemization, CRN: Cosmogenic Radionuclides, ESR: Electron Spin Resonance.

| Country | Location | Latitude | Longitude | Dating method | Confidence | Reference |
|---------|----------|----------|-----------|---------------|------------|-----------|
| Ecuador | Galera | 0.81 | -80.03 | IRSL | 5 | Pedoja et al., 2006 |
| Ecuador | Manta | -0.93 | -80.66 | IRSL, U/Th | 5 | Pedoja et al., 2006 |
| Ecuador | La Plata | -1.26 | -81.07 | U/Th | 5 | Pedoja et al., 2006 |
| Ecuador | Manta | -1.27 | -80.78 | IRSL | 5 | Pedoja et al., 2006 |
| Ecuador | Santa Elena | -2.21 | -80.88 | U/Th | 5 | Pedoja et al., 2006 |
| Ecuador | Puna | -2.60 | -80.40 | U/Th | 5 | Pedoja et al., 2006 |
| Peru | Cancas | -3.72 | -80.75 | Morphostratigraphy | 5 | Pedoja et al., 2006 |
| Peru | Mancora/Lobitos | -4.10 | -81.05 | Morphostratigraphy | 5 | Pedoja et al., 2006 |
| Peru | Talara | -4.56 | -81.28 | Morphostratigraphy | 5 | Pedoja et al., 2006 |
| Peru | Paita | -5.03 | -81.06 | Morphostratigraphy | 5 | Pedoja et al., 2006 |
| Peru | Bayovar/Illescas | -5.31 | -81.10 | IRSL | 5 | Pedoja et al., 2006 |
| Peru | Cerro Huevo | -15.31 | -75.17 | CRN | 5 | Saillard et al., 2011 |
| Peru | Chala Bay | -15.85 | -74.31 | CRN | 5 | Saillard, 2008 |
| Peru | Ilo | -17.55 | -71.37 | AAR | 5 | Ortlieb et al., 1996b; Hsu et al., 1989 |
| Chile | Punta Lobos | -20.35 | -70.18 | U/Th, ESR | 5 | Radtke, 1989 |
| Chile | Cobija | -22.55 | -70.26 | Morphostratigraphy | 4 | Ortlieb et al., 1995 |
| Chile | Michilla | -22.71 | -70.28 | AAR | 3 | Leonard & Wehmiller, 1991 |
| Chile | Hornitos | -22.85 | -70.30 | U/Th | 5 | Ortlieb et al., 1996 |
| Chile | Chacaya | -22.95 | -70.30 | AAR | 5 | Ortlieb et al., 1996 |
| Chile | Pampa Mejillones | -23.14 | -70.45 | U/Th | 5 | Victor et al., 2011 |
| Chile | SW Mejillones/Punta Jorge | -23.54 | -70.55 | U/Th, ESR | 3 | Radtke, 1989 |
| Chile | Coloso | -23.76 | -70.46 | ESR | 3 | Schellmann & Radtke, 1997 |
| Chile | Punta Piedras | -24.76 | -70.55 | CRN | 5 | Martinod et al., 2016 |
| Chile | Esmeralda | -25.91 | -70.67 | CRN | 5 | Martinod et al., 2016 |
| Chile | Caldera | -27.01 | -70.81 | U/Th, ESR | 5 | Marquardt et al., 2004 |
| Chile | Bahia Inglesa | -27.10 | -70.85 | U/Th, ESR | 5 | Marquardt et al., 2004 |
| Chile | Caleta Chanaral | -29.03 | -71.49 | CRN | 5 | Martinod et al., 2016 |
| Chile | Coquimbo | -29.96 | -71.34 | AAR | 5 | Leonard & Wehmiller, 1992; Hsu et al., 1989 |
| Chile | Punta Lengua de Vaca | -30.24 | -71.63 | U/Th | 5 | Saillard et al., 2012 |
| Chile | Punta Lengua de Vaca | -30.30 | -71.61 | U/Th | 5 | Saillard et al., 2012 |
| Chile | Quebrada Palo Cortado | -30.44 | -71.69 | CRN | 5 | Saillard et al., 2009 |
| Chile | Rio Limari | -30.63 | -71.71 | CRN | 5 | Saillard et al., 2009 |
| Chile | Quebrada de la Mula | -30.79 | -71.70 | CRN | 5 | Saillard et al., 2009 |
| Chile | Quebrada del Teniente | -30.89 | -71.68 | CRN | 5 | Saillard et al., 2009 |
| Chile | Puertecillo | -34.09 | -71.94 | IRSL | 5 | Jara-Munoz et al., 2015 |
| Chile | Pichilemu | -34.38 | -71.97 | IRSL | 5 | Jara-Munoz et al., 2015 |
| Chile | Putu | -35.16 | -72.25 | IRSL | 5 | Jara-Munoz et al., 2015 |
| Chile | Constitucion | -35.40 | -72.49 | IRSL | 5 | Jara-Munoz et al., 2015 |
| Chile | Constitucion | -35.44 | -72.47 | IRSL | 5 | Jara-Munoz et al., 2015 |
| Chile | Carranza | -35.58 | -72.61 | IRSL | 5 | Jara-Munoz et al., 2015 |
| Chile | Carranza | -35.64 | -72.54 | IRSL | 5 | Jara-Munoz et al., 2015 |
| Chile | Pelluhue | -35.80 | -72.54 | IRSL | 5 | Jara-Munoz et al., 2015 |
| Chile | Pelluhue | -35.80 | -72.55 | IRSL | 5 | Jara-Munoz et al., 2015 |
| Chile | Curanipe | -35.97 | -72.78 | IRSL | 5 | Jara-Munoz et al., 2015 |
| Chile | Arauco | -37.62 | -73.67 | IRSL | 5 | Jara-Munoz et al., 2015 |
| Chile | Arauco | -37.68 | -73.57 | CRN | 5 | Melnick et al., 2009 |
| Chile | Arauco | -37.71 | -73.39 | CRN | 5 | Melnick et al., 2009 |
| Chile | Arauco | -37.76 | -73.38 | CRN | 5 | Melnick et al., 2009 |
| Chile | Cerro Caleta Curi–anco | -39.72 | -73.40 | Tephrochronology | 4 | Pino et al., 2002 |
| Chile | South Curi–anco | -39.76 | -73.39 | Tephrochronology | 4 | Pino et al., 2002 |
| Chile | Valdivia | -39.80 | -73.39 | Tephrochronology | 4 | Pino et al., 2002 |
| Chile | Camping Bellavista | -39.85 | -73.40 | Tephrochronology | 4 | Pino et al., 2002 |
| Chile | Mancera | -39.89 | -73.39 | Tephrochronology | 5 | Silva, 2005 |

To quantify the reliability and consistency of our shoreline-angle measurements, we applied a quality rating from low
(1) to high (5) confidence. The four parameters that are included in our quality rating (QR) comprise a) the distance
to the nearest referencing point ($D_{RP}$), b) the confidence of the referencing point that is based on the dating method
used by previous studies ($C_{RP}$) (Pedoja et al., 2014), c) the measurement error in TerraceM ($E_T$), and (d) the pixel-
scale resolution of the topographic data set (R) (Fig. 3). We did not include the error that results from the usage of
different swath widths, since the calculated elevation difference with respect to the most frequently used 200 m swath
width is very low (< 0.5 m) (Fig. 2C). From the reference points we only used data points with a confidence value of
3 or greater (1 – poor, 5 – very good) based on the previous qualification of Pedoja et al. (2014). We further used this
confidence value to quantify the quality of the age constraints in the WALIS template. To account for the different
uncertainties of the individual parameters in the QR, we combined and weighted the parameters $D_{RP}$ and $C_{RP}$ in a first
equation claiming 60% of the final QR, $E_T$ in a second and R in a third equation weighted 30% and 10%, respectively.

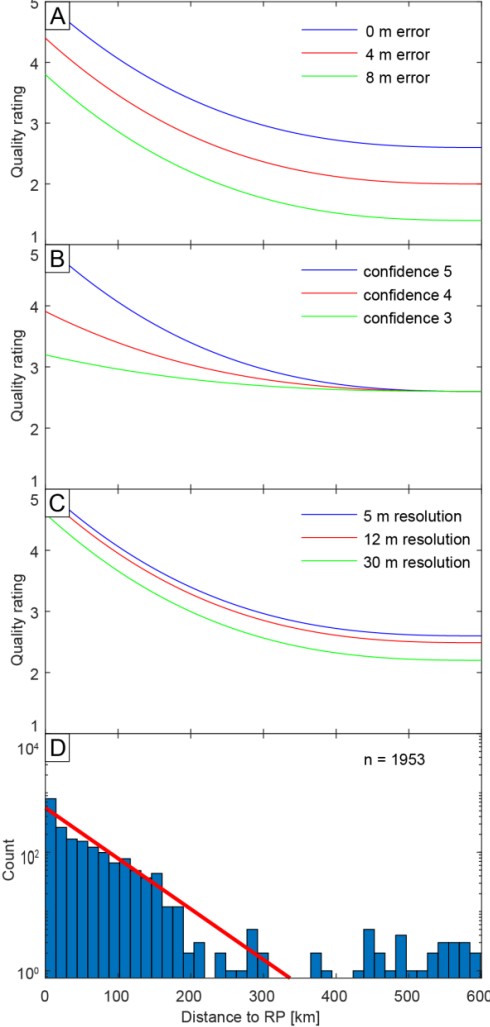

**Figure 3. Influence of the parameters on the quality rating. The y-axis is the distance to reference point (RP), x-axis is the quality rating, the color lines represent different values of quality rating parameters. While one parameter is being tested, the remaining parameters are set to their best values. That is why the QR does not reach values of 1 in the graphs displayed here. (A) Shoreline-angle elevation error. (B) Confidence value of the referencing point. (C) Topographic resolution of the DEM used for terrace-elevation estimation. (D) Histogram displaying the distribution of distances between each shoreline-angle measurement and its nearest RP (n: number of measurements). The red line is an exponential fit.**



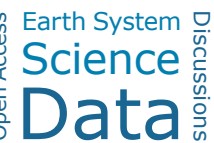

We justify these percentages by the fact that the distance and confidence to the nearest referencing point is of utmost
importance for identifying the MIS-5e terrace level. The measurement error represents how well the mapping of the
paleo-platform and paleo-cliff led to the shoreline-angle measurement, while the topographic resolution of the
underlying DEM only influences the precise representation of the actual topography and has little impact on
measurement itself. Furthermore, we added an exponent to the first part of the equation to reinforce low confidence
and/or high distance of the referencing point for low quality ratings. The following equation illustrates how we
calculated the individual parameters and the overall quality rating:
$$QR = 1 + 2.4 * \left( \frac{C_{RP}}{\max(C_{RP})} * \left( 1 - \frac{D_{RP}}{\max(D_{RP})} \right) \right)^e + 1.2 * \left( 1 - \frac{E_T}{\max(E_T)} \right) + 0.4 * 1.2 * \left( 1 - \frac{R}{\max(R)} \right)$$
The influence of each parameter to the quality rating can be observed in Fig. 3. We observe that for high $D_{RP}$ values
the QR becomes constant; likewise, the influence of QR parameters becomes significant for QR values higher than 3.
We justify the constancy of the QR for high $D_{RP}$ values (> 300 km) by the fact that most terrace measurements have
$D_{RP}$ values below 200 km (Fig. 3D). The quality rating is then used as a descriptor of the confidence of marine terrace-
elevation measurements.
**3.2.    Estimating coastal uplift rates**
Uplift-rate estimates from marine terraces *(u)* were calculated using the following equations:
$$\Delta H = H_T - H_{SL}$$
$$u = \frac{H_T - H_{SL}}{T}$$
where *ΔH* is the relative sea level, $H_{SL}$ is the sea-level altitude of the interglacial maximum, $H_T$ is the shoreline-angle
elevation of the marine terrace, and *T* its associated age (Lajoie, 1986).
We calculated the standard error *SE(u)* using the following equation from Gallen et al. (2014):
$$SE(u)^2 = u^2 \left( \left( \frac{\sigma_{\Delta H}^2}{\Delta H^2} \right) + \left( \frac{\sigma_T^2}{T^2} \right) \right)$$
where $\sigma_{\Delta H}^2$, the error in relative sea level, equals $(\sigma_{H_T}^2 + \sigma_{H_{SL}}^2)$. The standard-error estimates comprise the uncertainty
in shoreline-angle elevations from TerraceM ($\sigma_{H_T}$), error estimates in absolute sea level ($\sigma_{H_{SL}}$) from Rohling et al.
(2009), and an arbitrary range of 10 ka for the duration of the highstand ($\sigma_T$).
**3.3.    Tectonic parameters of the South American convergent margin**
We compared the deformation patterns of marine terraces along the coast of South America with proxies that included
crustal faults, bathymetric anomalies, trench-sediment thickness, and distance to the trench. To evaluate the possible





control of climatic parameters in the morphology of marine terraces, we compared our data set with wave heights,
tidal range, mean annual precipitation rate, and the azimuth of the coastline (Schweller et al., 1981; Bangs and Cande,
1997; von Huene et al., 1997; Collot et al., 2002; Ceccherini et al., 2015; Hayes et al., 2018; Santibáñez et al., 2019;
GEBCO Bathymetric Compilation Group, 2020) (Fig. 1).
To evaluate the potential correlations between tectonic parameters and marine terraces, we then analyzed the
latitudinal variability of these parameters projected along a curved "simple profile" and a 300-km-wide "swath profile"
following the trace of the trench. We used simple profiles for visualizing 2D data sets; for instance, to compare crustal
faults along the forearc area of the margin (Veloza et al., 2012; Melnick et al., 2020), we projected the seaward tip of
each fault. For the trench-sediment thickness, we projected discrete thickness estimates based on measurements from
reflection-seismic profiles of Bangs and Cande (1997), Collot et al. (2002), Huene et al. (1996), and Schweller et al.
(1981). Finally, we projected the discrete trench distances from the point locations of our marine terrace measurements
along a simple profile. To compare bathymetric features on the oceanic plate, we used a compilation of bathymetric
measurements at 450 m resolution (GEBCO Bathymetric Compilation Group, 2020). The data set was projected along
a curved 300-km-wide swath profile using TopoToolbox (Schwanghart and Kuhn, 2010).
Finally, to elucidate the influence of climatic factors on marine terrace morphology, we compared the elevation, but
also the number of measurements as a proxy for preservation and exposure of marine terraces. We calculated wave
heights, tidal ranges, and reference water levels at the point locations of our marine terrace measurements using the
Indicative Meaning Calculator (IMCalc) from Lorscheid and Rovere (2019). We used the maximum values of the
hourly significant wave height, and for the tidal range we calculated the difference between the highest and lowest
astronomical tide. The reference water level represents the averaged position of the paleo sea level with respect to the
shoreline-angle elevation and, together with the indicative range (uncertainty), quantifies the indicative meaning
(Lorscheid and Rovere, 2019). We furthermore used the high-resolution data set of Ceccherini et al. (2015) for mean
annual precipitation, and we compared the azimuth of the coast in order to evaluate its exposure to wind and waves.
To facilitate these comparisons, we extracted the values of all these parameters at the point locations of our marine
terrace measurements and project them along a simple profile. Calculations and outputs were processed and elaborated
using MATLAB® 2020b.
**4.    Results**
**4.1.    Marine terrace geomorphology and shoreline-angle elevations**
In the following sections we describe our synthesized database of last interglacial marine terrace elevations along the
WSC. Marine terraces of the last interglacial are generally well preserved and almost continuously exposed along the
WSC, allowing to estimate elevations with a high spatial density. To facilitate the descriptions of marine terrace-
elevation patterns, we divided the coastline into four sectors based on their main morphometric characteristics (Fig.
4): 1) the Talara bend in northern Peru and Ecuador, 2) southern and central Peru, 3) northern Chile, and 4) central
and south-central Chile. In total we carried out 1,843 MIS-5e terrace measurements with a median elevation of 30.1





m asl and 110 MIS-5c terrace measurements with a median of 38.6 m. The regions with exceptionally high marine
terrace elevations (≥ 100 m) comprise the Manta Peninsula in Ecuador, the San Juan de Marcona area in south-central
Peru, and three regions in south-central Chile (Topocalma, Carranza, and Arauco). Marine terraces at high altitudes
(≥ 60 m) can also be found in Chile on the Mejillones Peninsula, south of Los Vilos, near Valparaíso, in Tirua, and
near Valdivia, while terrace levels only slightly above the median elevation are located at Punta Galera in Ecuador,
south of Puerto Flamenco, at Caldera/Bahía Inglesa, near Caleta Chañaral, and near the Quebrada El Moray in the
Altos de Talinay area in Chile. In the next sections we described the characteristics of each site in detail, the names of
the sites are written in brackets following the same nomenclature as in the WALIS database.





**Figure 4. Shoreline-angle elevation measurements (colored points), referencing points (black stars), Quaternary faults (bold black lines) (Veloza et al., 2012; Melnick et al., 2020), and locations mentioned in the text for the four main morphometric segments (for location see Fig. 1A) (World Ocean Basemap: Esri, Garmin, GEBCO, NOAA NGDC, and other contributors). (A) Talara bend in Ecuador and northern Peru. (B) Central and southern Peru. (C) Northern Chile. (D) Central and south-central Chile.**

### 4.1.1.    Ecuador and northern Peru (1°N–6.5°S)

The MIS-5e terrace levels in Ecuador and northern Peru [sites Ec1 to Ec4 and Pe1] are discontinuously preserved along the coast (Fig. 5). They often occur at low elevations (between 12 m and 30 m) and show abrupt local changes in elevation, reaching a maximum at the Manta Peninsula. Punta Galera in northern Ecuador displays relatively broad and well-preserved marine terraces ranging between 40 and 45 m elevation and rapidly decreasing eastward to around 30 m asl across the Cumilínche fault [Ec1]. Farther south, between Pedernales and Canoa [Ec1], narrow terraces occur at lower altitudes of 22–34 m asl. A long-wavelength (~120 km) pattern in terrace-elevation change can be observed across the Manta Peninsula with the highest MIS-5e terraces peaking at ~100 m asl at its southern coast [Ec2]. This terrace level is hardly visible in its highest areas with platform widths smaller than 100 m due to deeply incised and narrowly spaced river valleys. We observe lower and variable elevations between 30 and 50 m across the Rio Salado fault in the San Mateo paleo-gulf in the north, while the terrace elevations increase gradually from ~40 m in the Pile paleo-gulf in the south [Ec3] toward the center of the peninsula (El Aromo dome) and the Montecristi fault [Ec3]. A lower terrace level correlated to MIS-5c displays similar elevation patterns as MIS-5e within the Pile paleo-gulf and northward. Near the Gulf of Guayaquil and the Dolores-Guayaquil megashear, the lowest terrace elevations occur at the Santa Elena Peninsula ranging between 17 and 24 m asl and even lower altitudes in its southern part, and on the Puna Island ranging between 11 and 16 m asl [Ec4]. In northern Peru [Pe1], we observe dismembered MIS-5e terraces in the coastal area between Cancas and Talara below the prominent Mancora Tablazo. "Tablazo" is a local descriptive name used in northern Peru (~3.5–6.5°S) for marine terraces that cover a particularly wide surface area (Pedoja et al., 2006b). South of Cancas, MIS-5e terrace elevations range between 17 and 20 m asl, reaching 32 m near Organos, and vary between 20 and 29 m in the vicinity of Talara. In the southward continuation of the Talara harbor, the Talara Tablazo widens, with a lower marine terrace at around 23 m asl immediately north of Paita Peninsula reaching 30 m asl in the northern part of the peninsula. The last occurrence of well-preserved MIS-5e terraces in this sector is at the Illescas Peninsula, where terrace elevations decrease from around 30 m to 17 m asl southward.



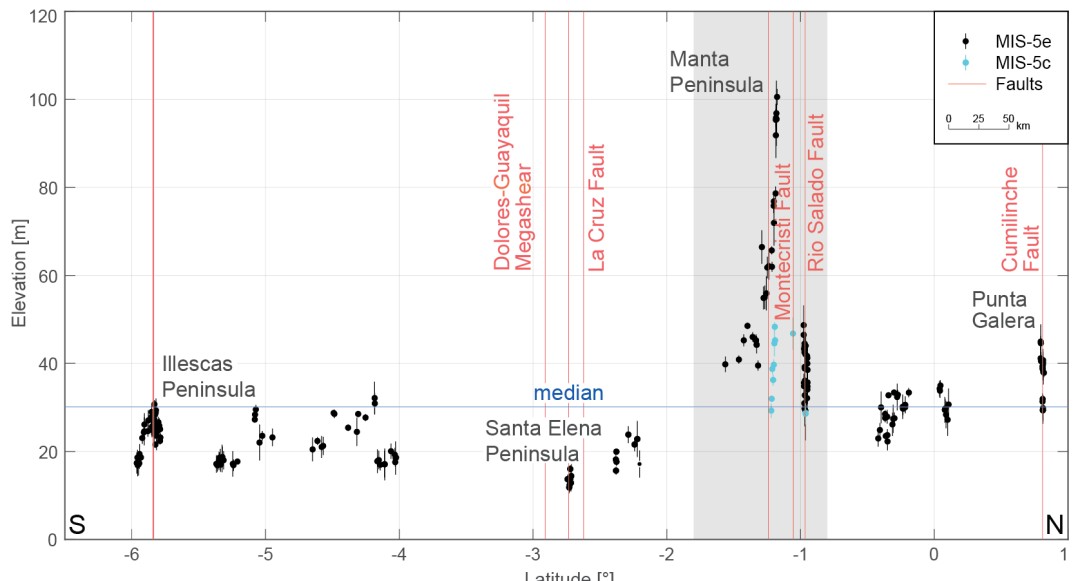

**Figure 5. Measured shoreline-angle elevations of MIS-5e and 5c in Ecuador and northern Peru. A high and inferred long-**
**wavelength change in terrace elevation occurs at the Manta Peninsula (gray area) and quite low altitudes farther south at**
**the Santa Elena Peninsula. Several short-scale terrace-elevation changes coincide with faulting at Punta Galera and on the**
**Illescas Peninsula. Median elevation: 30.1 m. For location see Fig. 4A.**
### 4.1.2. Central and southern Peru (6.5°–18.3°S)
This segment comprises marine terraces at relatively low and constant elevations, but which are rather discontinuous
[sites Pe2 to Pe10], except in the San Juan de Marcona area, where the terraces increase in elevation drastically (Fig.
6). The coast in north-central Peru exhibits poor records of MIS-5e marine terraces, characterized by mostly narrow
and discontinuous remnants that are sparsely distributed along the margin. Marine terraces increase in elevation from
11 to 35 m asl south of Chiclayo [Pe2] and decrease to 17 m asl near Cercado de Lima [Pe3, Pe4], forming a long-
wavelength (~600 km), small amplitude (~20 m) upwarped structure. The MIS-5e terrace levels are better expressed
in the south-central and southern part of Peru at elevations between 35 and 47 m asl in San Vicente de Cañete,
decreasing to approximately 30 m asl in the vicinity of Pisco [Pe5]. South of Pisco, the coastal area becomes narrow
with terrace elevations ranging between 25 and 34 m asl [Pe6] and increasing abruptly to 74–79 m near Puerto Caballas
and the Río Grande delta. MIS-5e terrace elevations are highest within the San Juan de Marcona area, reaching 109–
93 m at Cerro Huevo and 87–56 m at Cerro Trés Hermanas [Pe7]. These higher terrace elevations coincide with a
wider coastal area, a better-preserved terrace sequence, and several crustal faults, such as the San Juan and El Huevo
faults.
Terrace heights west of Yauca indicate a further decrease to 50–58 m before a renewed increase to 70–72 m can be
observed in the Chala embayment [Pe8]. We observe a similar trend in elevation changes for the shoreline angles
attributed to the MIS-5c interglacial within the previously described high-elevationarea: 31–39 m near the Río Grande
delta, 62–58 m below the Cerro Huevo peak, 64–27 m below the Cerro Trés Hermanas peak [Pe7], 36–40 m near
Yauca, and 34–40 m within the Chala embayment [Pe8]. Besides various changes in between, terrace elevations
decrease slowly from 54 m south of the Chala region to 38 m near Atico [Pe8]. The overall decrease south of the San
Juan de Marcona area therefore contrasts strikingly with the sharper decrease to the north. These high-elevation marine
terraces, which extend ~250 km along the coast from north of the San Juan de Marcona area to south of Chala Bay,
constitute one of the longest wavelength structures of the WSC. Southeast of Atico, less well-preserved marine terraces
appear again in form of small remnants in a narrower coastal area. Starting with elevations as low as 24 m, MIS-5e
terrace altitudes increase southeastward to up to 40 m near Mollendo [Pe9], before they slightly decrease again. The
broader and quite well-preserved terraces of the adjacent Ilo area resulted in a smooth increase from values greater
than 25 m to 33 m and a sudden decrease to as low as 22 m across the Chololo fault [Pe9]. North of the Arica bend,
shoreline-angle measurements yielded estimates of 24–29 m in altitude [Pe10].

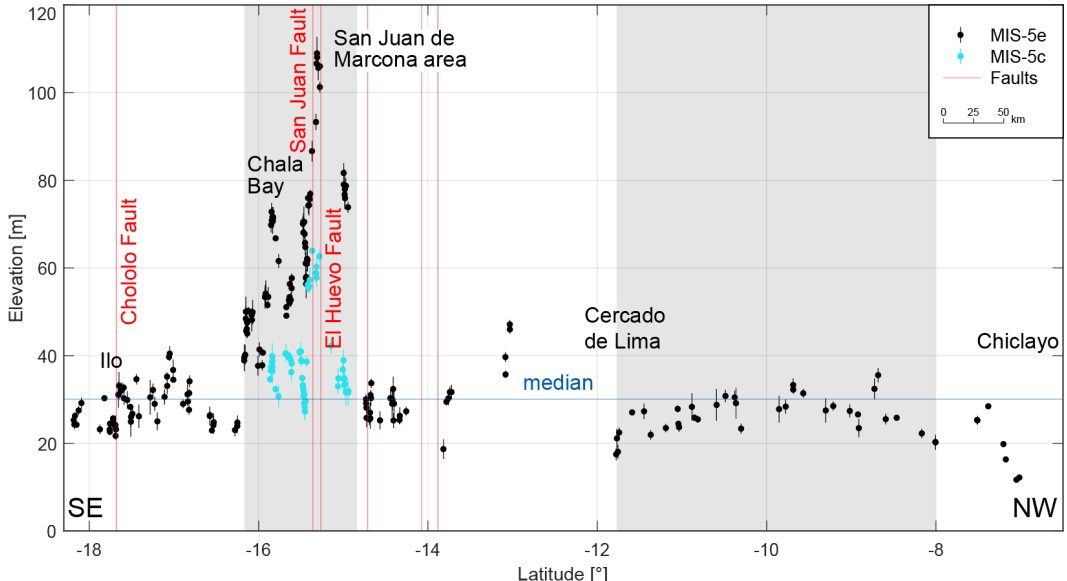

**Figure 6. Measured shoreline-angle elevations of MIS-5e and 5c terraces in central and southern Peru. While only sparsely**
**preserved terraces below the median (30.1 m) occur in central Peru between Chiclayo and Lima, a relatively broad and**
**asymmetric distribution of marine terraces characterizes the area of San Juan de Marcona. For location see Fig. 4B.**
**4.1.3.    Northern Chile (18.3°–29.3°S)**
Along the northern Chilean coast, marine terraces of the MIS-5e are characterized by a variable elevation pattern and
the occurrence of numerous crustal faults associated with the Atacama fault system, although the changes in terrace
elevation are not as pronounced as in the northern segments (Fig. 7) [sites Ch1 to Ch7]. The local widening of the
coastal area near the Arica bend narrows southward with MIS-5e terraces at elevations of between 24 and 28 m asl in
northernmost Chile [Ch1]. Just north of Pisagua, we measured shoreline-angle elevations of well-preserved marine
terraces between 19 and 26 m across the Atajana fault [Ch1]. A short-scale zigzag pattern starting with shoreline-
angle elevation values of 32 m south of Iquique and south of the Zofri and Cavancha faults decreases rapidly to
approximately 22 m, increases again to similar altitudes and drops as low as 18 m toward Chanabaya south of the
Barranco Alto fault [Ch1]. A gentle, steady rise in terrace elevations can be observed south of Tocopilla where
altitudes of 25 m are attained. South of Gatico, terrace markers of the MIS-5e highstand increase and continue
northward for much of the Mejillones Peninsula within an approximate elevation range of 32–50 m asl, before reaching
a maximum of 62 m asl at the Pampa de Mejillones [Ch2]. With its ~100 km latitudinal extent, we consider this
terrace-elevation change to be a medium-wavelength structure. Although no MIS-5e terrace levels have been
preserved at the Morro Mejillones Horst (Binnie et al., 2016), we measured shoreline-angle elevations at the elevated
southwestern part of the peninsula that decrease sharply from 55 to 17 m asl in the vicinity of the Mejillones fault
system [Ch2]. After a short discontinuation of the MIS-5e terrace level at Pampa Aeropuerto, elevations remain
relatively low between 19–25 m farther south [Ch2]. Along the ~300-km coastal stretch south of Mejillones, marine
terraces are scattered along the narrow coastal area ranging between 25 and 37 m asl [Ch3]. South of Puerto Flamenco,
MIS-5e terrace elevations range between 40 and 45 m asl until Caldera and Bahía Inglesa [Ch4]. The MIS-5e marine
terrace elevations decrease abruptly south of the Caldera fault and the Morro Copiapó (Morro Copiapó fault) to
between 25 and 33 m asl, reaching 20 m asl north of Carrizal Bajo [Ch4]. In the southernmost part of the northern
Chilean sector, the MIS-5e terraces rise from around 30 m asl to a maximum of 45 m asl near the Cabo Leones fault
[Ch4], before declining abruptly near Caleta Chañaral and Punta Choros [Ch5, Ch6, Ch7].

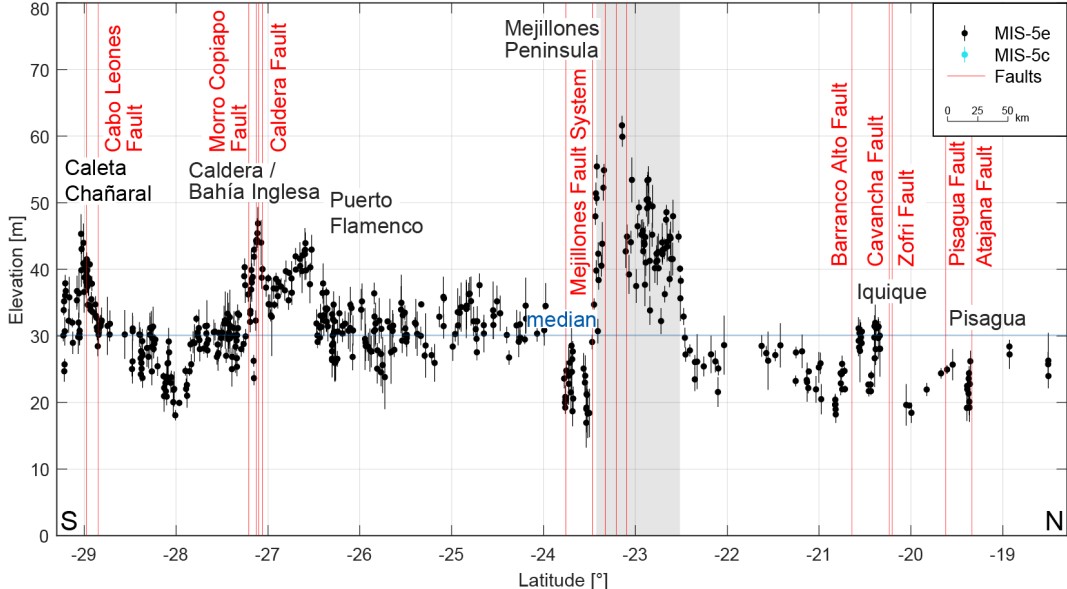

**Figure 7. Measured shoreline-angle elevations of MIS-5e and 5c terraces in northern Chile. Faults as well as asymmetrically**
**uplifted marine terraces of up to 60 m elevation characterize the Mejillones Peninsula, reaching values below 20 m at the**
**southern margin. Terrace elevations attain peak values south of Puerto Flamenco, at Caldera/Bahía Inglesa, and north of**
**Caleta Chañaral, while in between minimum altitudes below 20 m occur (north of Carizal Bajo). Median elevation is 30.1**
**m. For location see Fig. 4C.**
**4.1.4.    Central Chile (29.3°–40°S)**
Marine terraces along central Chile display variable, high-amplitude terrace-elevation patterns associated with
numerous crustal faults, and include a broad-scale change in terrace altitudes with the highest MIS-5e marine terrace



elevations of the entire South American margin on the Arauco Peninsula (Fig. 8) [sites Ch8 to Ch78]. South of Punta
Choros, marine terrace elevations decrease from values close to 40 to 22 m asl north of Punta Teatinos [Ch8, Ch9]. A
maximum elevation of 40 m is reached by the terraces just south of this area [Ch10] whereas north of La Serena, a
sharp decrease leads to values between 20 and 30 m for marine terraces south of Coquimbo Bay and in the Tongoy
Bay area [Ch11, Ch12]. South of Punta Lengua de Vaca, our measurements of the exceptionally well-preserved
staircase terraces are in the same elevation range between 20 and 30 m, increasing slowly to 40 m near the Quebrada
el Moray [Ch13]. Although we could not observe a significant change in terrace elevation across the Puerto Aldea
fault, we measured an offset of ~7 m across the Quebrada Palo Cortado fault. MIS-5e terrace levels decrease thereafter
and vary between 20 and 30 m in altitude until north of Los Vilos [Ch14–Ch18], where they increase in elevation
[Ch19], reaching 60 m near the Rio Quilimari [Ch20]. The marine terraces become wider in this area and are associated
with scattered sea stacks. Decreasing farther south to only 20 m asl [Ch21–Ch25], the coastal area narrows and has
terrace heights of up to 64 m near Valparaíso that are cut by numerous faults (e.g., Valparaíso and Quintay faults)
[Ch26–Ch32]. Another low-elevation area follows southward, with values as low as 17 m [Ch33–Ch35]. Farther south,
between 34°S and 38°S, broad (~200 km at Arauco), medium (~45 km at Topocalma), and narrow (Carranza)
upwarped zones occur that are manifested by variable terrace elevations. These include prominent high-terrace
elevations at Topocalma with a maximum of 180 m [Ch36–Ch39], slightly lower levels of 110 m at Carranza [Ch42–
Ch47], exceptionally low values near the Río Itata (< 10 m) [Ch48–Ch64, Ch66], and the most extensive and highest
shoreline-angle elevations on the Arauco Peninsula with elevations in excess of 200 m [Ch67–Ch73]. Additionally,
we measured MIS-5c terrace elevations in the three higher exposed areas with a range of 20–55 m at Carranzo, and a
few locations at Topocalma (76–81 m) and Arauco (117–123 m). The medium-wavelength structure of Topocalma is
bounded by the Pichilemu and Topocalma faults, and near Carranza several fault offsets (e.g., Pelluhue and Carranza
faults) are responsible for the short-wavelength changes in terrace elevation. In contrast, crustal faulting is nearly
absent in the high-elevation and long-wavelength structure at Arauco. MIS-5e terrace elevations are highly variable
within a short area south of the Arauco Peninsula near the Tirua fault, increasing rapidly from 27 m to 78 m and
decreasing thereafter to around 20 m [Ch74, Ch75]. The continuity of terraces is interrupted by the absence of terrace
levels between Río Imperial and Río Toltén, but resumes afterward with a highly frequent zigzag pattern and multiple
faults (e.g., Estero Ralicura and Curinanco faults) from as low as 18 m to a maximum of 40 m [Ch76, Ch77]. In this
area locations with the highest terrace levels comprise the terraces near Mehuín and Calfuco. A final increase in
shoreline-angle elevations from around 20–30 m up to 76 m near Valdivia coincides with the southern terminus of our
terrace-elevation measurements [Ch78].

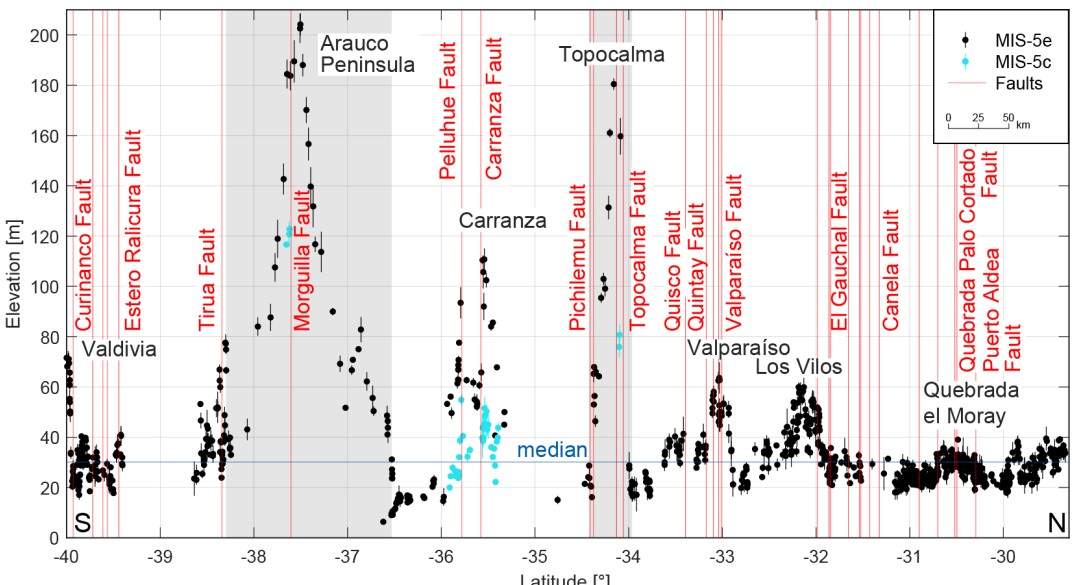

**Figure 8. Measured shoreline-angle elevations of MIS-5e and 5c terraces in central Chile. Extensive faulting coincides with**
**various high terrace elevations of the last interglacial highstand north of Los Vilos, near Valparaiso, at Topocalma,**
**Carranza, and near Valdivia. The most pronounced and long-wavelength change in terrace elevation occurs on the Arauco**
**Peninsula with maximum elevations over 200 m and minimum elevations below 10 m north of Concepción. Median**
**elevation: 30.1 m. For location see Fig. 4D.**
**4.2.    Statistical analysis**
Our statistical analysis of mapped shoreline-angle elevations resulted in a maximum kernel density at 28.96 m with a
95% confidence interval from 18.59 m to 67.85 m ($2\sigma$) for the MIS-5e terrace level (Fig. 9A). The MIS-5c terrace
yielded in a maximum kernel density at a higher elevation of 37.20 m with $2\sigma$ ranging from 24.50 m to 63.92 m. It is
important to note that the number of MIS-5c measurements is neither as high nor as continuous as compared to that
of the MIS-5e level. MIS-5c data points were measured almost exclusively in sites where MIS-5e reach high elevations
(e.g., San Juan de Marcona with MIS-5e elevations between 40 and 110 m).
The distribution of measurement errors was studied using probability kernel-density plots for each topographic
resolution (1-5 m LIDAR, 12 m TanDEM-X, and 30 m TanDEM-X). The three data sets display similar distributions
and maximum likelihood probabilities (MLP); for instance, LiDAR data show a MLP of 0.93 m, the 12 m TanDEM-
X a MLP of 1.16 m, and 30 m TanDEM-X a MLP of 0.91 m (Fig. 9B). We observe the lowest errors from the 30 m
TanDEM-X, slightly higher errors from the 1-5 m LiDAR data, and the highest errors from the 12 m TanDEM-X.
This observation is counterintuitive as we would expect lower errors for topographic data sets with higher resolution.
The reason for these errors is probably related to the higher number of measurements using the 12 m TanDEM-X
(1564) in comparison with the measurements using 30 m TanDEM-X (50), which result in a higher dispersion (Fig.
9B). In addition, the relation between terrace elevations and error estimates shows that comparatively higher errors
are associated with higher terrace elevations, although the sparse point density of high terrace-elevation measurements
prevents a clear correlation from being recognized (Fig. 9C).

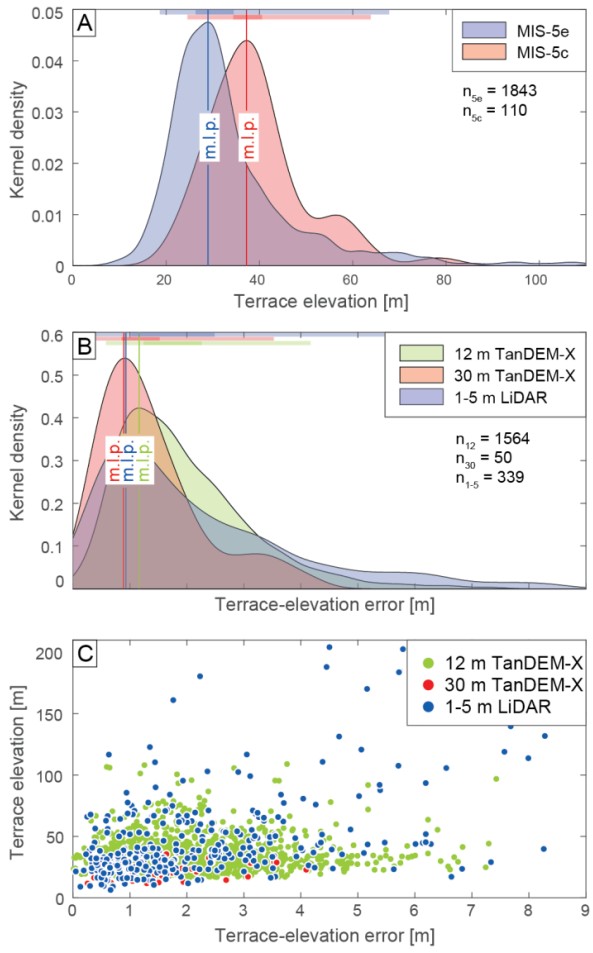

**Figure 9. Statistical analysis of measured shoreline-angle elevations. (A) Kernel-density plot of MIS-5e and 5c terrace elevations with maximum likelihood probabilities (MLP.) at 28.96 m elevation for MIS-5e and 37.20 m elevation for MIS-5c (n: number of measurements). Colored bars on top highlight the standard deviations σ and 2σ. (B) Kernel-density and their associated standard-deviation (σ and 2σ) calculations of terrace-elevation errors for source DEMs of various resolutions. The most abundant 12 m TanDEM-X has a MLP-error of 1.16 m, while the 30 m TanDEM-X and the 1-5 m LIDAR produce slightly lower errors of 0.91 m and 0.93 m, respectively. (C) Terrace-elevation errors plotted against terrace elevation for the individual source DEMs. Although the point density for high terrace elevations is low, a weak correlation of high errors with high terrace elevations can be observed.**

### 4.3. Coastal uplift-rate estimates

We calculated uplift rates from 1953 terrace-elevation measurements of MIS-5e (1843) and MIS-5c (110) along the South American margin with a median uplift rate of approximately 0.22 m/ka (Fig. 10). As with the distribution of terrace elevations, we similarly observed several short-scale and long-scale high-amplitude changes in uplift rate along the coast. The most pronounced long-wavelength highs (≥ 1° latitude) in uplift rate are located on the Manta Peninsula (0.79 m/ka), in the San Juan de Marcona area (0.85 m/ka), and on the Arauco Peninsula (1.62 m/ka). Medium-wavelength structures include the Mejillones Peninsula (0.47 m/ka) and Topocalma (1.43 m/ka), while shorter wavelength structures that are characterized by exceptionally high uplift rates seem to be limited to the central Chilean part of the coastline, especially between 31.5° and 40°S. The most striking example includes Carranza with an uplift rate of up to 0.87 m/ka since the formation of the oldest MIS-5 terrace levels. Lower, but still quite high, uplift rates were calculated for areas north of Los Vilos (0.46 m/ka), near Valparaíso (0.49 m/ka), and near Valdivia (0.59 m/ka). The lowest uplift rates along the South American margin occur at Penco immediately north of Concepción (0.03





m/ka), south of Chiclayo in northern Peru (0.07 m/ka), and on the southern Santa Elena Peninsula in Ecuador (0.07
m/ka).

## 5.   Discussion

### 5.1.   Advantages and limitations of the database of last interglacial marine terrace elevations along the WSC

In this study we generated a systematic database of last interglacial marine terrace elevations with unprecedented
resolution based on an almost continuous mapping of ~2,000 measurements along 5,000 km of the WSC. This opens
up several possibilities for future applications in which this database can be used; for example, the fact that marine
terraces are excellent strain markers used in studies on deformation processes at regional scale, comparisons between
deformation rates at different temporal scales or analyses linking specific climate-driven coastal processes, landscape
evolution and tectonics. However, there are a number of limitations and potential uncertainties that can affect the use
of this database in such studies without taking several caveats into consideration.
One of the most critical limitations of using the database is associated with the referencing points used to tie our
marine terrace measurements, which are in turn based on the results and chronological constraints provided by
previous studies. The referencing points are heterogeneously distributed along the WSC, resulting in some cases of
up to 600 km distance to the nearest constrained point, such as in Central Peru [e.g. Pe2]. This may have a strong
influence on the confidence in the measurement of the marine terrace elevation at these sites. In addition, the
geochronological control of some of the referencing points may be based on dating methods with pronounced
uncertainties (e.g., amino acid racemization, electron spin resonance, cosmogenic radionuclides), which may result in
equivocal interpretations and chronologies of marine terrace levels. In order to address these potential factors of
uncertainty we defined a quality rating (see section 3.1.), which allows classifying our mapping results based on their
confidence and reliability. Therefore, by considering measurements above a defined quality it is possible to increase
the confidence level of future studies using this database; however, this might result in a decrease of the number of
measurement points available for analysis and comparison.

### 5.2.   Tectonic and climatic controls on the elevation and morphology of marine terraces along the WSC

In this section we provide a brief synthesis of our data set and its implications for coastal processes and overall
landscape evolution that are driven by a combination of tectonic and climatic characteristics. This synthesis
emphasizes the significance of our comprehensive data set for a variety of coastal research problems that were briefly
introduced in section 5.1. Our detailed measurements of marine terraces along the WCS reveal variable elevations and
a heterogeneous distribution of uplift rates associated with patterns of short-, medium-, and long-wavelengths. In
addition, we observe different degrees of development of marine terraces along the margin expressed in variable
shoreline-angle density. There are several possible causes for this variability, which we explore by comparing terrace-
elevation patterns with different climatic and tectonic parameters.

### 5.2.1.   Tectonic controls on coastal uplift rates





The spatial distribution of the MIS-5 marine terrace elevations along the convergent South American margin has
revealed several high-amplitude and long-wavelength changes with respect to tectonically controlled topography.
Long-wavelength patterns in terrace elevation (~$10^2$ km) are observed at the Manta Peninsula in Ecuador, central Peru
between Chiclayo and Lima, San Juan de Marcona (Peru), and on the Arauco Peninsula in Chile, while medium-
wavelength structures occur at Mejillones Peninsula and Topocalma (Chile). Instead, short-wavelength patterns in
MIS-5 terrace elevations are observed for instance near Los Vilos, Valparaíso, and Carranza in Chile.
The subduction of bathymetric anomalies has been shown to exert a substantial influence on upper-plate deformation
(Fryer and Smoot, 1985; Taylor et al., 1987; Macharé and Ortlieb, 1992; Cloos and Shreve, 1996; Gardner et al., 2013;
Wang and Bilek, 2014; Ruh et al., 2016), resulting in temporally and spatially variable fault activity, kinematics, and
deformation rates (Mann et al., 1998; Saillard et al., 2011; Morgan and Bangs, 2017; Melnick et al., 2019). When
comparing the uplift pattern of MIS-5 marine terraces and the bathymetry of the oceanic plate, we observe that the
two long-wavelength structures in this area, on the Manta Peninsula and in the San Juan de Marcona, both coincide
with the location of the subducting Nazca and Carnegie ridges, respectively (Fig. 10A and B); this was also previously
observed by other authors (Gutscher et al., 1999; Pedoja et al., 2006a; Saillard et al., 2011). In summary, long-
wavelength structures at the coast may be associated with deep-seated processes (Melosh and Raefsky, 1980; Watts
and Daly, 1981) possibly related to changes in the mechanical behavior of the plate interface. In this context it is
interesting that the high uplift rates on the Arauco Peninsula do not correlate with bathymetric anomalies, which may
suggest a different deformation mechanism. The scarcity of crustal faults described in the Arauco area rather suggests
that shallow structures associated with crustal bending and splay-faults occasionally breaching through the upper crust
(Melnick et al., 2012; Jara-Muñoz et al., 2015; Jara-Muñoz et al., 2017; Melnick et al., 2019) cause long-wavelength
warping and uplift there (Fig. 10A).
In contrast, small-scale bathymetric anomalies correlate in part with the presence of crustal faults perpendicular to the
coastal margin near, for instance, the Juan Fernandez, Taltal, and Copiapó ridges (Fig. 10B), which result in short
wavelength structures and a more localized differentiation of uplifted terraces. This emphasizes also the importance
of last interglacial marine terraces with respect to currently active faults, which might be compared in the future with
short-term deformation estimates from GPS or the earthquake catalog. In summary, short-wavelength structures may
be associated with crustal faults that root at shallower depths within the crust (Jara-Muñoz et al., 2015; Jara-Muñoz et
al., 2017; Melnick et al., 2019).
The thickness of sediment in the trench is an additional controlling factor on forearc architecture that may determine
which areas of the continental margin are subjected to subduction erosion or accretion (Hilde, 1983; Cloos and Shreve,
1988; Menant et al., 2020). Our data shows that the accretionary part of the WSC (south of the intersection with the
Juan Fernandez Ridge at 32.9°S) displays faster median uplift rates of 0.26 m/ka than in the rest of the WSC (Fig. 10B
and C). However, no clear correlation is observed between trench fill, uplift rates, and the different structural patterns
in the erosive part of the margin. On the other hand, we observe lower uplift rates for greater distances from the trench
at the Arica bend, in central Peru, and in the Gulf of Guayaquil, while higher uplift rates occur in areas closer to the
trench, such as near the Nazca and Carnegie ridges and the Mejillones Peninsula.



### 5.2.2. Climatic controls on the formation and preservation of last interglacial marine terraces

The latitudinal climate differences that characterize the western margin of South America may also control coastal morphology and the generation and preservation of marine terraces (Martinod et al., 2016b). In order to evaluate the influence of climate in the generation and/or degradation of marine terraces, we compared the number of marine terrace measurements, which is a proxy for the degree of marine terrace preservation, and climatically controlled parameters such as wave height, tidal range, coastline orientation, and the amount of precipitation.

The maximum wave height along the coast of South America decreases northward from ~8 to ~2 m (see section 3.3, Fig. 10D). Similarly, the tidal range decreases progressively northward from 2 to 1 m between Valdivia and San Juan de Marcona, followed by a rapid increase to 4 m between San Juan de Marcona and the Manta Peninsula. We observe an apparent correlation between the number of measurements and the tidal range in the north, between Illescas and Manta (Fig. 10F). Likewise, the increasing trend in the number of measurements southward matches with the increase in wave height (Fig. 10D). An increase of wave height and tidal range may lead to enhanced erosion and morphologically well-expressed marine terraces, which is consequently reflected in a higher number of measurements. Furthermore, we observe low values for the reference water level (< 0.7 m) resulting from tide and wave-height estimations in IMCalc (Lorscheid and Rovere, 2019), which are used to correct our shoreline-angle measurements (see section 3.3.).

The control of wave-erosion processes on the morphological expression of marine terraces may be counteracted by erosional processes such as river incision. We note that the high number of preserved marine terraces between Mejillones and Valparaíso decreases southward, which coincides with a sharp increase in mean annual precipitation from 10 to 1000 mm/yr (Fig. 10E and F). However, in the area with a high number of measurement points between the Illescas Peninsula and Manta we observe an opposite correlation: higher rainfall associated with an increase of marine terrace preservation (Fig. 10E). This anticorrelation suggests that the interplay between marine terrace generation and degradation processes apparently buffer each other, resulting in different responses under different climatic conditions and coastal settings.

The higher number of marine terraces between Mejillones and Valparaíso and north of Illescas corresponds with a SSW-NNE orientation of the coastline (azimuth between 200 and 220°). In contrast, NW-SE to N-S oriented coastlines (azimuth between 125 and 180°), such as between the Arica and Huancabamba bends, correlate with a lower number of marine terrace measurements (Fig. 10E and F). This observation is, however, counterintuitive considering that NW-SE oriented coastlines may be exposed more directly to the erosive effect of storm waves associated with winds approaching from the south. We interpret the orientation of the coastline therefore to be of secondary importance at regional-scale for the formation of marine terraces compared to other parameters, such as wave height, tidal range, or rainfall.

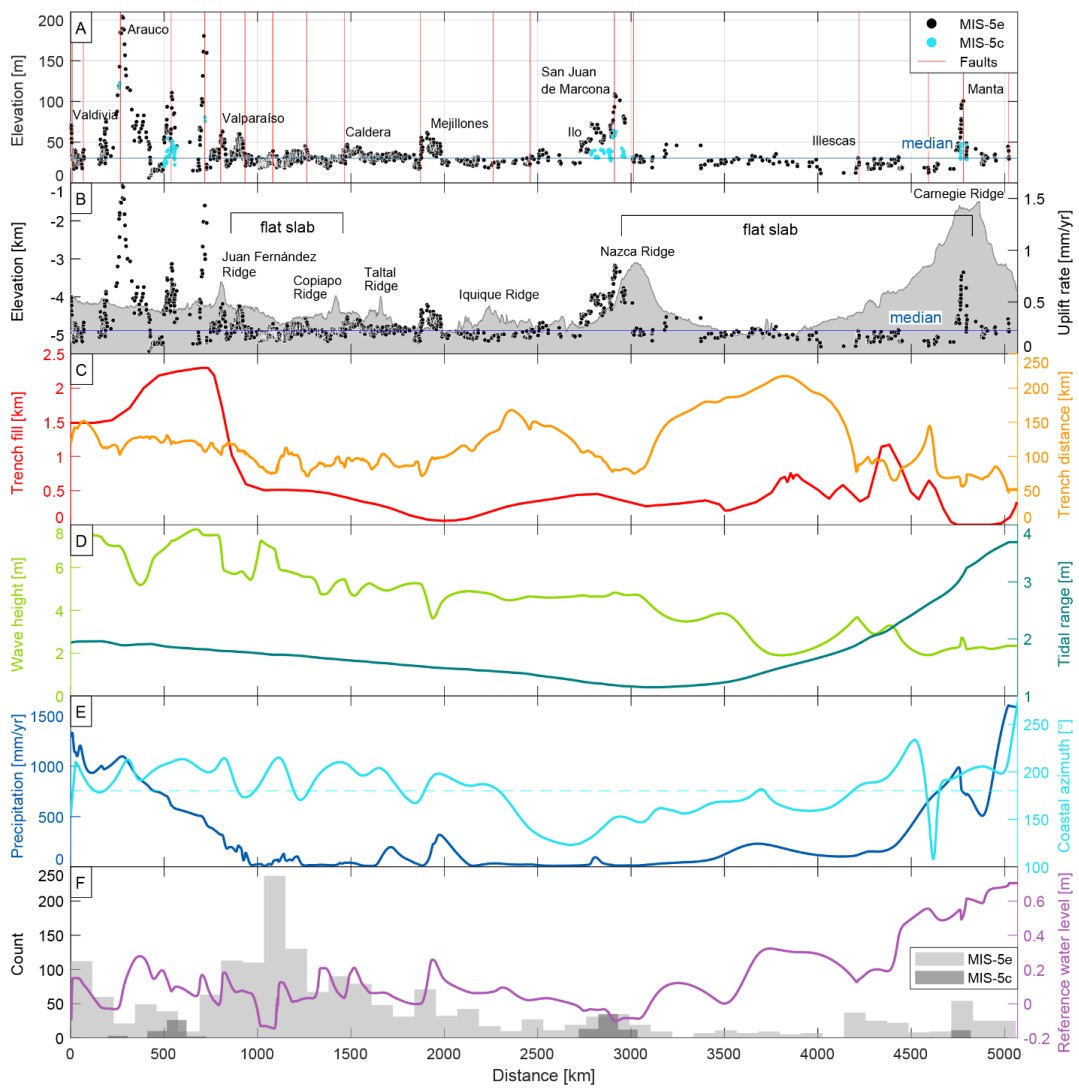

**Figure 10. Terrace-elevation and uplift-rate estimates plotted in comparison with various parameters (i.e., bathymetry,**
**trench fill, trench distance, wave height, tidal range, precipitation, and coastal azimuth) that might influence the disparate**
**characteristics of our marine terrace distribution that has been revealed by our data set. We projected these parameters,**
**elevations, and uplift rates to a S-N-oriented polyline that represents the trench. (A) Terrace-elevation measurements and**
**most important crustal faults (Veloza et al., 2012; Melnick et al., 2020). This shows the range of altitudes in different regions**
**along the coast and possible relationships to crustal faulting. The blue horizontal line indicates the median elevation (30.1**
**m). (B) Coastal uplift rates and mean bathymetry (GEBCO Bathymetric Compilation Group, 2020) of a 150-km swath west**
**of the trench. The blue horizontal line indicates the median uplift rate (0.22 mm/a). (C) Sediment thickness of trench-fill**
**deposits (red) (Bangs and Cande, 1997) and the distance of the trench from our terrace measurements (orange). Flat-slab**
**segments of the subducting Nazca plate are indicated for central Chile and Peru. (D) Maximum wave heights along the**
**WSC (light green) and the tidal range (dark green) between highest and lowest astronomical tides (Lorscheid and Rovere,**
**2019). (E) Precipitation (blue) along the WSC (Ceccherini et al., 2015) and azimuthal orientation of the coastline (cyan). (F)**
**Histogram of terrace-elevation measurements along the WSC.**



## 6. Conclusions

We measured 1,953 shoreline-angle elevations as proxies for paleo-sea levels of the MIS-5e and 5c terraces along ~5,000 km of the WSC between Ecuador and Southern Chile. Our measurements are based on a systematic methodology and the resulting data have been standardized within the framework of the WALIS database. Our mapping was tied using referencing points based on previously published terrace-elevation estimates and age constraints that are summarized in the compilation of Pedoja et al. (2014). The limitations of this database are associated with the temporal accuracy and spatial distribution of the referencing points, which we attempt to consider by providing a quality rating value to each measurement. The marine terrace elevations display a median value of 30.1 m for the MIS-5e level and a median uplift rate of 0.22 m/ka for MIS-5e and 5c. The lowest terrace elevations and uplift rates along the entire South American margin occur immediately north of Concepción in Chile (6 m, 0.03 m/ka), south of Chiclayo in northern Peru, and on the Santa Elena Peninsula in Ecuador (both 12 m, 0.07 m/ka). The regions with exceptionally high marine terrace elevations ($\geq$ 100 m) comprise the Manta Peninsula in Ecuador, the San Juan de Marcona area in south-central Peru, and three regions in south-central Chile (Topocalma, Carranza, and Arauco).

The pattern of terrace elevations displays short-, medium- and long-wavelength structures controlled by a combination of various mechanisms. Long-wavelength structures may be controlled by deep-seated processes at the plate interface, such as the subduction of major bathymetric anomalies (e.g. Manta Peninsula and San Juan de Marcona region). In contrast, short-wavelength deformation patterns may be controlled by crustal faults rooted within the upper plate (e.g., between Mejillones and Valparaíso).

Latitudinal climate characteristics along the WSC may influence the generation and preservation of marine terraces. An increase in wave height and tidal range generally results in enhanced erosion and morphologically well-expressed, sharply defined marine terraces, which correlates with the southward increase in the number of our marine terrace measurements. Conversely, river incision and lateral scouring in areas with high precipitation may degrade marine terraces, thus decreasing the number of potential marine terrace measurements, such as observed south of Valparaíso.

*Data availability*. The South American database of last interglacial shoreline-angle elevations is available online at http://doi.org/10.5281/zenodo.4309748 (Freisleben et al., 2020). The description of the WALIS-database fields can be found at https://doi.org/10.5281/zenodo.3961543 (Rovere et al., 2020).

*Author contributions*. The main compilers of the database were R.F., J.M.M., and J.J. The paper was written by R.F. with significant input from J.J., D.M., M.S. regarding interpretation and further improvements of graphical data representation.

*Acknowledgments*. We thank Alessio Rovere for his assistance with the WALIS database. The WALIS database was developed by the ERC Starting Grant "Warmcoasts" (ERC-StG-802414) and PALSEA. PALSEA is a working group of the International Union for Quaternary Sciences (INQUA) and Past Global Changes (PAGES), which in turn received support from the Swiss Academy of Sciences and the Chinese Academy of Sciences. The structure of the database was designed by A. Rovere, D. Ryan, T. Lorscheid, A. Dutton, P. Chutcharavan, D. Brill, N. Jankowski, D. Mueller, M. Bartz, E.J. Gowan and K. Cohen. This study was supported by the Millennium Scientific Initiative of the Chilean government through grant NC160025 "Millennium Nucleus CYCLO The Seismic Cycle Along Subduction Zones", Chilean National Fund for Development of Science and Technology (FONDECYT) grants 1181479 and





1190258, the ANID PIA Anillo ACT192169. R.F. was supported by a research grant of Deutsche
Forschungsgemeinschaft to M.S. (DFG STR373/41-1).

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
