# Peer review of "Marine terraces of the last interglacial period along the Pacific"

_Earth System Science Data, 2020_

## Referee Comment (RC1) · Vincent Regard (Referee) · 12 Jan 2021

The manuscript entitled "Marine terraces of the last interglacial period along the Pacific coast of South America (1°N-40°S)" presents a supposedly exhaustive dataset of the marine terraces of the last interglacial all along the Pacific margin of the Andes (between 1°N and 40°S)". The dataset is interesting and deserves publication. Nevertheless, the manuscript deserves more attention as a number of points could be strengthened to make it more comprehensive.

I begin with a very general remark. Like the authors, I have often tried to make studies that are as objective as possible. But when dealing with data from Nature, we are often

forced to make subjective choices. Here, it concerns the location of the profiles, or the parameters of the QR equation. I would be grateful if the authors could further explain how they choose the location of the profiles and possibly show a less "obvious" site than Chala. For QR, one understands, one accepts, the weight put on each of the terms, but the exponent e seems a bit magical. Couldn't the authors get rid of e?

Another general remark: the paper is difficult to follow because many points are badly presented or discussed. Firstly, how can the authors be sure of the age of the terraces? Is the signal continuous (which I doubt since there are spaces of more than 100 km between some terraced areas in Figure 4)? For example, I was not aware of any terraces in north-central Peru, which I thought were subsiding (see le Roux et al. 2000): are the terraces really MIS 5? Secondly, the authors use terminology that I understand established by their own group. More explanations/details would be necessary about what the indicative meaning is, about what a referencing point is, about the location and nomenclature of the measuring points (figure 4 should show the names of the points, Pe2, Ch1.... As well as some of the names in figures 5 - 8).

The TerraceM data is clean. In order to interpret them in terms of uplift rate or ancient sea level, the authors try to precisely quantify the uplift. This is not trivial: the authors do a good job on sea level but ignore the fact that the current shoreline angle is not at an altitude of 0, it is often higher. Even if the authors do not account for this offset, it would be good if they mentioned it and possibly the uncertainty it introduces.

Nevertheless, the systematic use of TerraceM is a good initiative, and I support the publication of this data with the paper that goes with it.

Specific comments

Lines 104-105. Steep vs flat slab not really introduced.

Paragraph 2.1.2. I understand the interest of presenting active tectonics, but the paragraph is neither concise nor exhaustive, so I doubt its usefulness. Perhaps it would

be better to quote the two recent compilations by Melnick et al (2019) and Costa et al (2020).

Lines 325-335. Explanation difficult to follow. Why not starting with the QR equation?

Lines 575-581. It is indeed interesting; generations of researchers have not waited for the authors to get interested in using marine terraces to study the uplift of key areas such as Arauco or Mejillones.

Lines 582-593. I am much more interested in less studied areas such as north-central Peru. The authors could expand a little more on this point.

Technical corrections

Line 40: Reference to Siddall OK, but references to marine terraces would be welcome.

Line 55. I would move the reference to Regard et al. to line 50 as it represents a fairly continuous signal.

Line 80 and in other places, as line 288. The reference may be Pedoja et al. 2011, more focused on the last interglacial than Pedoja et al. 2014.

Line 184: "a slight increase in distance". Which distance?

Line 187: "Wave erosion forms wave-cut terrace levels" This is what the community think, but it is not certain (see Premaillon et al. 2018).

Lines 281-282. I think I understand, but the sentence "The DEMs were converted to orthometric heights using the ellipsoid projection of the World Geodetic System (WGS1984) and the EGM2008 (EEGM08) geoid" is misspelled. Does this mean that the authors used a grid of EGM heights above WGS84?

Legend of Figure 3: There was a switch between x-axis and y-axis.

QR. Indicate that it varies between 1 and 5; there is an error on the 3rd coefficient: 0.4 rather than 0.4 * 1.2.

Line 555. Is there a reason why error and dispersion are correlated (not clear to me)?

Line 617. The authors must quote Macharé and Ortlieb, a key paper.

Line 651-652. A ref to support this assertion is missing.

Line 667-671. It is possible that wave power is not the main driver for coastal erosion...

Additional references

Costa, C. et al., 2020, Hazardous faults of South America; compilation and overview: Journal of South American Earth Sciences, v. 104, p. 102837, doi:10.1016/j.jsames.2020.102837.

le Roux, J.P., Correa, C.T., and Alayza, F., 2000, Sedimentology of the Rimac-Chillon alluvial fan at Lime, Peru, as related to Plio-Pleistocene sea-level changes, glacial cycles and tectonics: Journal Of South American Earth Sciences, v. 13, p. 499–510.

Prémaillon, M., Regard, V., Dewez, T.J.B., and Auda, Y., 2018, GlobR2C2 (Global Recession Rates of Coastal Cliffs): a global relational database to investigate coastal rocky cliff erosion rate variations: Earth Surface Dynamics, v. 6, p. 651–668, doi:https://doi.org/10.5194/esurf-6-651-2018.

---

## Referee Comment (RC2) · Paula Marques Figueiredo (Referee) · 1 Feb 2021

The manuscript entitled "Marine terraces of the last interglacial period along the Pacific coast of South America (1◦ N–40◦ S)" by Roland Freisleben and co-authors presents an enormous amount of research and work, putting forward an exhaustive compilation of data regarding the Last Interglacial Marine Terraces for a significant part of the Western South American coastline. Therefore, I believe this to be an interesting manuscript to be published and I congratulate the authors for this effort. The application of the TerraceM methodology is most useful for this area and will provide relevant information that has not yet been fully characterized.

[Figure]

However, there is room for improvement and I strongly encourage the authors to submit a revised final manuscript. I present a list of comments and minor corrections/ suggestions, and there are two main aspects that I would like to see properly addressed, especially in a manuscript dedicated to the last interglacial.

The authors only provide the example of Chala Bay (southern Peru) to illustrate their methodology. Chala Bay is probably a good example, since it exhibits clear morphology and has geochronology information. The authors identified the two terraces as MIS 5e and MIS 5c, and based their interpretation in Saillard (2008) data. Taking in consideration the inner edges elevations for the two terraces, and assuming them as MIS 5e and MIS 5c, the calculation of the uplift rates for each is different, implying uplift rates changes (using Sidall et al., 2007 and Creveling et al., 2017 for example). However, if we consider that the lower terrace is not MIS 5c but instead MIS 5a, then the uplift rate for both marine terraces is very similar, which is a plausible configuration. In fact, when looking at Figure 2A, it is possible to note that between the paleo-shorelines inferred for the two terraces illustrated, a third paleo-shoreline is easily visible between the swath profile in 74°16W and 74°17W and in close to the profile immediately south from the larger drainage in the Bay area. This other possible inner edge is thus located between the two paleo-shorelines presented. This immediately suggests that the less preserved morphology between the terraces can correspond to the MIS 5c and that the inner-edges at ∼70±2 m and ∼37±2 may instead be respectively MIS 5e and MIS 5a. A MIS 5c evidence, is only present when there were conditions to promote its preservation, since in most cases worldwide was indeed re-occupied by MIS 5a. Testing if possible less preserved feature is evidence of MIS 5c, can be performed for example, by estimating uplift rates and look for consistency with MIS 5e and 5a uplift rates. This is a relevant question for most MIS 5 users and needs to be clarified.

Saillard (2008) in fact suggests the same, that the morphology could be a mix of MIS 5c and/or MIS 5a and ultimately favoring a MIS 5a interpretation: - "Dans la zone de Chala‐Tanaka‐Chaviña, la terrasse marine à +60 m, dans la baie de Chala, a

un âge supérieur à 100 ka, et correspond vraisemblablement au SI 5 et au SSI 5c ou a (Figure 84). La terrasse marine à +94 m, dans la baie de Chala, a un âge minimum de 220 ka et correspond au SI 7 et au SSI 7c. De même, la terrasse marine à +154 m dans la baie de Chala, a un âge minimum de 250 ka et correspondrait donc au SI 9 et au SSI 9a. La terrasse d'abrasion marine à +90 m, au niveau du Cap de Tanaka, a un âge de 171 ka ± 21 ka et correspond au SI 7 et au SSI 7a (page 202, Saillard 2008)" - "Nous avons corrélé la terrasse +60 m au SSI 5e (122 ± 7 ka), conformément à Goy et al. (1992) et à nos âges 10Be obtenus sur un cône alluvial déposé sur la terrasse, indiquant un âge supérieur à 100 ka. En effet, la terrasse +60 m est le premier niveau étendu et séparé de la terrasse supérieure par un grand escarpement qui mesure ∼30 m. . . . A partir des corrélations de Goy et al. (1992), nous avons corrélé les terrasses marines inférieures +2 m, +11 m et +31 m, à l'Holocène moyen (SI 1 ; 6000 ± 1000 ans), au SSI 3 et au SSI 5a (Table 8)." (page 208, Saillard, 2008).

Note that the elevations of Saillard are not the same presented in this manuscript, which I think is a consequence of presenting TerraceM data (and maybe orthometric elevation), and this difference should in fact be checked. The MIS 5c or MIS 5a is a most relevant question. Why have the authors chosen to consider the morphology as MIS 5c? Is there any other evidence that supports it? As geochronology data? This also raises questions about the other locations where the authors also interpret MIS 5c landforms. If MIS 5c was recognized (and generally less preserved worldwide) why was it not possible to identify MIS 5a? If that was the case, what mechanisms differ from 112ka to 80 ka? And why is the Quality Rating for this location rated as high confidence?

Difficulties in characterizing MIS 5 and younger landforms bring me to the second major question. GIA causes a known impact to crustal vertical deformation that needs to be taken into account to infer paleo sea level positioning and decode tectonic imprint (e.g. Creveling et al., 2015, Simms et al., 2016). Comments regarding possible Glacio-Isostatic-Adjustments effects in the elevation of last interglacial and paleo sea-levels

are absent from this manuscript. It is likely that theses effects along the Western South America coast may vary from null or very little to a certain amount, however, this has not been introduced and discussed and the reader is not aware if it was considered or excluded and if the annexed dataset took those effects in consideration.

List of comments:

Line 38 – is there any difference between abrasion platforms and marine terraces? If so, can you explain what abrasion platforms are? Are marine terraces depositional landforms and abrasion platforms erosional? Line 60 – again what is the difference between marine terrace sequences and abrasion platforms? Clarifying may be useful for the reader. Maybe add definition of rasas, as well. Clarify what are the chosen nomenclatures for this manuscript. Lines 63-65 – "The marine terrace morphology comprises a gently inclined marine abrasion platform or depositional surface that terminates landward at a steeply sloping paleo-cliff surface. The intersection point between both surfaces represents the sea-level position during the formation of the marine terrace also known as shoreline angle" Please consider that it represents the higher sea-level associated to this landform and that it there may be a discrepancy between the timing of the shoreline angle trimming and the length of the MIS

Line 65 – please introduce the terminology Inner Edge, since it will be used in the later part of the manuscript and never explained.

Line 112 – "Several bathymetric anomalies have been recognized on the subducting Nazca plate." The bathymetric anomalies are landforms at the bathymetric surface, rather than on the plate. Please rephrase, explaining the landform, and if the landforms are in the continental shelf, abyssal plain? Why are they anomalies? Are they reliefs standing it out from a background? Are they seamounts or depressions, since both cases can be bathymetric anomalies? Since you will debate later that these features when subducted may cause an impact, consider describing their sizes, how high and wide are they (elevation from the ocean floor). Maybe include lithologies if you know

(are they volcanoes, structural reliefs?)

112-113 – "most prominent anomalies being subducted beneath South America are the Carnegie and Nazca" the anomalies are not subducted per se, but rather the morphologies "

Line 114 – "The Carnegie Ridge subducts roughly parallel with the convergence direction and its position should have remained relatively stable beneath the continent" please rephrase the word position and continent. The position is relative and it changes through time, consider using geometry, shape and for continent, consider renaming it with the concept of plate.

Line 119-120 "these bathymetric anomalies are thought to influence the characteristics of interplate coupling and seismic rupture" please consider rephrasing to something like these larger reliefs (seamounts etc? explain what they are) previously present at the surface of the now subducted plates are thought to . . ..

Line 124 and 126 – "amount of sediments" please consider rephrasing to volume of sediments, since amount could refer of number of different type of sediments, number of clasts. . . etc.

Line 127-128 –"thick trench sediments" consider rephrasing to thick trench sediment sequences. Is not sufficiently clear.

Line 141 – consider replacing dominates with is dominant or is prevalent.

Line 152 – "Smaller coastal fault systems" how small? You gave one order of magnitude for one example (2000 km) and now the reader doesn't know if small can be 800, 400, 50 km. . .

Line 139-157 – These paragraphs describe a lot of places that are not mentioned in figure 1 and the reader doesn't really know their location. In addition, although not the scope per se of this manuscript, consider adding a sentence of two, stating that the different kinematics reflect the deformation (different areas/ segments) associated

to the subduction. Also, can you add reference to a figure? For someone who is unfamiliar with the Geography, this is hard to follow.

Line 169-185 – These paragraphs describe a lot of places that are not mentioned in figure 1 and the reader doesn't really know their location, as for example Gulf of Guayaquil (3°S) and the Dolores-Guayaquil megashear, San Juan de Marcona area.

Based on comments for lines 139-157 and 169-185, would it be possible to add another map to Fig 1 with more information regarding sites and in land structures kinematics (suggestion, different colors for different kinematics)? It is very hard to read the small numbers referring to the location names.

Line 188 – what is rasa in comparison with abrasion platforms? Clarifying the different nomenclatures is very positive.

Line 222 – can you define what the Maule segment is? This nomenclature is not included in the map, and it has not been introduced earlier.

Line 228 – "extensive uplift rate of 0.31 to 0.42 m/ka", Can you add information for the period of time corresponding to this uplift rate, since MIS 5, since before? Add a time period.

Line 229 – 0.34 m/ka is not lower than 0.31 to 0.42 m/ka, is actually within the range. Can you rephrase, please?

Line 230 – 0.17– 0.21 m/ka for what time period?

Line 231-235 – "Marine Terraces above . . . last 125 ka according to.." this sentence is confusing, maybe rewrite to clarify. Consider adding the word between to help differentiate.

Line 243 – "oldest Pleistocene shore platforms" can you provide information regarding the age?

Line 243- 246 – "The Central Andean rasa (15°–33°S) and the oldest Pleistocene

shore platforms – which are also generally wider – indicate accelerated and spatially continuous uplift after a period of tectonic stability or subsidence. According to Melnick (2016), the Central Andean rasa has experienced slow and steady long-term uplift at $0.13 \pm 0.04$ m/ka during the Quaternary," It is my understanding from what is written that the Central Andean rasa were subjected to a low uplift rate, favoring their development. Is this long-term uplift at $0.13 \pm 0.04$ m/ka still the same? Was the period of tectonic stability or subsidence prior to the Central Andean rasa? Or the uplift initiated already during the Quaternary, and if yes, when? This sentence can be improved for clarification. Remember that today and yesterday are Quaternary as well.

Comment to the entire section 2.2.2 – Would greatly benefit if supported by an additional figure with geographic and some structural as suggested earlier. The reader is not aware of what is the Maule segment for example. It is very hard to follow locations.

Line 299 – Please verify if the provided link is correct

Line 299 – "placed swath profiles of variable width perpendicular to the previously mapped inner edge" It has not been given sufficient explanation regarding the selection of sites to study and where to place the swath profiles. How many for region, etc. . I understand their spacing being also controlled by drainage density, but maybe a sentence here to add further information is helpful.

Line 304 – "show only minimal deviations of less than 0.5 m" is this deviation regarding vertical or horizontal? Please clarify.

Line 308 – "we measured 1843 and 110 shoreline-angle elevations" in a total of 1953? It seems that a word is missing after 1843. What is the difference between 1843 and 110?

Line 309 – "5e and MIS-5c terrace levels". How do you know that it is MIS 5c and not MIS 5a? As far as I understand, there is not a direct numerical dating for it. Please discuss this identification as MIS 5c.

Figure 2 – This figure is extremely important, by being taken as an example of the methodology for the entire area. Due to its relevance, I have some comments. The shaded relief is not very visible, and the color scheme is not necessary, also because you have information for the inner edges elevations along the swath profiles. Slope information is barely noticeable and probably not necessary.

It lacks relevant information, most needed to validate your MIS interpretations. One can observe several other terraces higher and older that the one with the inner-edge at ∼70±2 m elevation. Those terraces are in fact the ones that were dated with Terrestrial Cosmogenic Nuclides by Saillard (2008) and providing a time constrain for the marine terrace interpretation. I was unable to find in the thesis a direct TCN numerical dating for MIS 5e in Chala Bay, but rather for MIS 7 and older, and I think the authors need to clarify this information in the manuscript.

I strongly advise to add to this image the inner edges from previous terraces (as dashed lines for instances) and add locations of TCN samples with numerical ages ( X± x ka) . I also question the sudden change elevation of the lower inner edge in the western most and southern most profiles. Is it a steep section of the cliff? Why in one drives the inner edge up and in the other lowers it? Why was not possible to measure the upper terrace one? Since the confidence level given to these coastal landforms as MIs 5e and MIS 5c is 5 (high confidence), a proper justification supported in evidences is required.

Figure 3D – is n= 1953 referring to what? terrace-elevation measurements ? What is the 1843 referred earlier?

Line 342- 343 – "The following equation illustrates how we calculated the individual parameters and the overall quality rating:" Besides identifying variables, the authors do not provide any explanation for the equation per se, and it would be most adequate to do so. Can you please provide the basis for this equation? Maybe including references as well.

[Figure]

Line 397-398 – Only now is explained what 1843 and 110 are. Please consider explaining this earlier. (line 308).

Figures 5 to 8 – should probably be organized together, similar to what was done with figure 4. It would be easier to compare the different regions and then it will be easier to compare that figure with figure 4. Consider including location of the ridges previously mentioned as anomalies.

Line 542 "Our statistical analysis of mapped shoreline-angle elevations" for the entire area? Does it make sense, when there are so many tide and wave height variations, not to mention changes imposed by tectonics?

Line 551-556 – "We observe the lowest errors from the 30 m TanDEM-X, slightly higher errors from the 1-5 m LiDAR data, and the highest errors from the 12 m TanDEM-X. This observation is counterintuitive as we would expect lower errors for topographic data sets with higher resolution. The reason for these errors is probably related to the higher number of measurements using the 12 m TanDEM-X (1564) in comparison with the measurements using 30 m TanDEM-X (50), which result in a higher dispersion (Fig. 9B)." These sentences will greatly benefit from a better explanation or rewriting. It is unclear if the authors are stating that 30 m TAnDEM-X is a better topographic dataset for this type of analysis, or if their results of error analysis result from their own sampling.

Line 627-629 "This emphasizes also the importance of last interglacial marine terraces with respect to currently active faults, which might be compared in the future with short-term deformation estimates from GPS or the earthquake catalog." Short term estimates from GPS are really very short in time and may not reflect long term rates expressed by the faults, and active structures may even be seismic silent for Holocene times. Maybe focus in discuss their validity as a Late Pleistocene marker for crustal vertical deformation.

Line 651-652 – "An increase of wave height and tidal range may lead to enhanced

erosion and morphologically well-expressed marine terraces, which is consequently reflected in a higher number of measurements" This statement probably needs further evidence.

Line 664- 669 – has the lithology effect been considered? Some lithologies are easier to trim that others and other are more easily eroded. Has structural control (bedding, etc) taken into account in this analysis? Has the cliff erosion and in particular the Holocene retreat been compared? Obviously a higher cliff retreat will erode terraces which don't necessarily mean that they were never formed.

Line 693-694 "The marine terrace elevations display a median value of 30.1 m for the MIS-5e level and a median uplift rate of 0.22 m/ka for MIS-5e and 5c." Why presenting median values in such a dynamic and variable tectonic setting along 5000 km? Is it relevant as a conclusion?

References added:

Creveling, J. R., Mitrovica, J. X., Hay, C. C., Austermann, J., and Kopp, R. E., 2015, Revisiting tectonic corrections applied to Pleistocene sea-level highstands: Quaternary Science Reviews, v. 111, p. 72-80.

Creveling, J.R., Mitrovica, J.X., Clark, P.U., Waelbroeck, C., Pico, T., 2017. Predicted bounds on peak global mean sea level during marine isotope stages 5a and 5c, Quaternary Science Reviews, V. 163, 193-208, https://doi.org/10.1016/j.quascirev.2017.03.003.

Simms, A. R., Rouby, H., and Lambeck, K., 2016, Marine terraces and rates of vertical tectonic motion: The importance of glacio-isostatic adjustment along the Pacific coast of central North America: Geological Society of America Bulletin, v. 128, p. 81-93.

**Cerro El Huevo**

| Terrasses | Hsu (1988a et b) SI | Age SI (ka) | Ortlieb et Macharé (1990) SI | Age SI (ka) | Ce travail Age $^{10}$Be (ka) | SI | Age SI (ka) |
|---|---|---|---|---|---|---|---|
| +41 m | 3 | 60 | 3 | 60 | | 5a | 85 |
| +56 m | | | 5a | 85 | | | |
| +72 m | 5e | 125 | 5c | 105 | | 5c | 105 |
| +105 m | 7a | 200 | 5e | 125 | | 5e | 122 |
| +150 m | 9 | 300 | 7a | 195 | 228 ± 28 | 7e | 232,5 |
| +170 m | 11 | 400 | 7c | 220 | | 9a | 280 |
| +190 m | 13 | 500 | 9c | 330 | 318 ± 37 | 9c | 321 |
| +220 m | | | | | 400 ± 49 | 11 | 405 |

**Cerro Tres Hermanas**

| Terrasses | Ortlieb et Macharé (1990) SI | Age SI (ka) | Ce travail SI | Age $^{10}$Be (ka) | SI | Age SI (ka) |
|---|---|---|---|---|---|---|
| +41 m | 5a | 85 | | | 5a | 85 |
| +55 m | 5c | 105 | | | 5c | 105 |
| +80 m | 5e | 125 | | | 5e | 122 |
| +131 m | 7a | 195 | | | 7e | 232,5 |
| +145 m | 7c | 220 | | | 7a | 195 |
| +162 m | 9c | 330 | | 353 ± 10 | 9c | 321 |
| +177 m | | | | | 11 | 405 |

**Chala**

| Terrasses | Goy et al. (1992) SI | Age SI (ka) | Ce travail Age $^{10}$Be (ka) | SI | Age SI (ka) |
|---|---|---|---|---|---|
| +2 m | 1 | 0,5 | | 1 | 0,5 |
| +11 m | 3 | 60 | | 3 | 60 |
| +31 m | 5a | 85 | | 5a | 85 |
| +60 m | 5e | 125 | <100 | 5e | 122 |
| +94 m | 7 | ~215 | >220 | 7a | 197 |
| +109 m | | | | 7e | 232,5 |
| +154 m | 9 | ~300 | >250 | 9c | 321 |
| +160 m | | | | | |
| Entre +178 m et +212 m | 11 | 400 | | | |

**Ilo**

| Terrasses | Ortlieb et al. (1996) SI | Age SI (ka) | Ce travail Age $^{10}$Be (ka) | SI | Age SI (ka) |
|---|---|---|---|---|---|
| +25 m | 5e | 125 | | 5e | 122 |
| +40 m | 7 | ~215 | | 7e | 232,5 |
| +80 m | 9 | ~300 | 300 ± 30 | 9c | 321 |
| +120 m | | | | 11 | 405 |

Table 8 : Table récapitulative des corrélations des différents niveaux de terrasses proposées dans les études antérieures et dans ce travail, pour les zones de San Juan de Marcona (Cerros El Huevo et Tres Hermanas), Chala et Ilo. SI : Stade isotopique.

**Fig. 1.**

---

## Author Comment (AC1) · 12 Mar 2021

**Comments by Reviewer #1: Vincent Regard**

I begin with a very general remark. Like the authors, I have often tried to make studies that are as objective as possible. But when dealing with data from Nature, we are often forced to make subjective choices. Here, it concerns the location of the profiles, or the parameters of the QR equation. I would be grateful if the authors could further explain how they choose the location of the profiles and possibly show a less "obvious" site than Chala.

> We included more information on the placement of swath profiles in lines 321-323, and a second site at Punta Galera in Ecuador to show the placement of swath profiles and sensitivity tests to swath width.

For QR, one understands, one accepts, the weight put on each of the terms, but the exponent e seems a bit magical. Couldn't the authors get rid of e?

> We included the exponent to adjust the QR value according to the observed natural distribution of distances from referencing points, which follows an exponential relationship (Fig. 4D). For clarity, we extended the explanation of the exponential term in text (lines 399-400).

Another general remark: the paper is difficult to follow because many points are badly presented or discussed. Firstly, how can the authors be sure of the age of the terraces? Is the signal continuous (which I doubt since there are spaces of more than 100 km between some terraced areas in Figure 4)? For example, I was not aware of any terraces in north-central Peru, which I thought were subsiding (see le Roux et al. 2000): are the terraces really MIS 5?

> To account for the potential uncertainties in the age of a terrace level, we use the distance to the nearest referencing point and the confidence of its age constraint, both of which are included in the QR. We extended the explanation of these topics in lines 298 and 301.
>
> Le Roux et al. (2000) proposed subsidence to have occurred during the Pliocence and early Pleistocene. However, it is difficult to extrapolate this subsidence to Late Pleistocene time scales. Furthermore, the study of Le Roux et al. (2000) is a local analysis that cannot be easily extrapolated to regional scales. Other authors such as Macharé and Ortlieb (1992) proposed steady-state conditions or subsidence based on qualitative observations. We are aware that terrace mapping is challenging in north-central Peru and referencing points are less dense compared to other areas farther south; however, our estimates account for the uncertainties related to reference-point distance in the quality rating.

Secondly, the authors use terminology that I understand established by their own group. More explanations/details would be necessary about what the indicative meaning is, about what a referencing point is, about the location and nomenclature of the measuring points (figure 4 should show the names of the points, Pe2, Ch1.... As well as some of the names in figures 5 - 8).

> The indicative meaning is taken from Lorscheid and Rovere (2019) and consists of the range between the lower and upper limit of sea-level formation – the indicative range – as well as its estimated position – the reference water level. Instead, "referencing points" are previously published terrace heights and age constraints. For clarity, we extended the explanations on these terms in lines 333-336 and 298-301, respectively.
>
> We added a short explanation of the abbreviations in the text (lines 472-473) and in the caption of Fig. 5 (lines 478-479), and we followed the reviewer's suggestion of adding the nomenclature in Fig. 6-9.

The TerraceM data is clean. In order to interpret them in terms of uplift rate or ancient sea level, the authors try to precisely quantify the uplift. This is not trivial: the authors do a good job on sea level but ignore the fact that the current shoreline angle is not at an altitude of 0, it is often higher. Even if the authors do not account for this offset, it would be good if they mentioned it and possibly the uncertainty it introduces. Nevertheless, the systematic use of TerraceM is a good initiative, and I support the publication of this data with the paper that goes with it.

> To account for the uncertainty between current shoreline-angle elevation and sea-level position, we used the indicative meaning (Fig. 11F, WALIS database, lines 74-76, 333-336 and 724-726). The indicative meaning consists of the range between the lower and upper limit of sea-level formation – the indicative range – as well as its mathematically averaged position, which corresponds to the reference water level (Lorscheid and Rovere, 2019). All these values are incorporated in the WALIS database and the reference water level is included in the discussion figure (Fig. 11). We changed the corresponding description in the methods (lines 333-336).

***Specific comments***

Lines 104-105. Steep vs flat slab not really introduced.

> Flat and steeper subduction angles are now introduced in lines 99-106.

Paragraph 2.1.2. I understand the interest of presenting active tectonics, but the paragraph is neither concise nor exhaustive, so I doubt its usefulness. Perhaps it would be better to quote the two recent compilations by Melnick et al (2019) and Costa et al (2020).

> We are convinced that an introduction to the major faults and fault systems is important because we refer to them later in the results and discussion sections (see also Fig. 6-9, Fig. 11). We added more information to this paragraph to make it clearer and added the references indicated (lines 141-147).

Lines 325-335. Explanation difficult to follow. Why not starting with the QR equation?

> We agree and rearranged this part and added some information to make it easier to understand (lines 375-400).

Lines 575-581. It is indeed interesting; generations of researchers have not waited for the authors to get interested in using marine terraces to study the uplift of key areas such as Arauco or Mejillones. Lines 582-593. I am much more interested in less studied areas such as north-central Peru. The authors could expand a little more on this point.

> The point of this study is to review previous studies and providing an almost continuous and methodologically uniform terrace mapping. Our study allows for different regions along the South American margin to be compared. Terrace-elevation estimates are denser and more precise in areas with well-preserved terraces and age constraints from previous studies. Information on marine terraces in north-central Peru can be found in the results section (4.1.2.). Their less-well developed morphology and sparse availability of age constraints is reflected in the quality rating.

***Technical corrections***

Line 40: Reference to Siddall OK, but references to marine terraces would be welcome.

> We included two more references.

Line 55. I would move the reference to Regard et al. to line 50 as it represents a fairly continuous signal.

*We moved the reference Regard et al. (2010) from line 55 to line 50.*

Line 80 and in other places, as line 288. The reference may be Pedoja et al. 2011, more focused on the last interglacial than Pedoja et al. 2014.

*We changed the references from Pedoja et al. (2014) to Pedoja et al. (2011).*

Line 184: "a slight increase in distance". Which distance?

*The increase in coast-trench distance. We added this to line 190.*

Line 187: "Wave erosion forms wave-cut terrace levels" This is what the community think, but it is not certain (see Premaillon et al. 2018).

*We briefly mention this ongoing discussion and added this reference to lines 196-197.*

Lines 281-282. I think I understand, but the sentence "The DEMs were converted to orthometric heights using the ellipsoid projection of the World Geodetic System (WGS1984) and the EGM2008 (EEGM08) geoid" is misspelled. Does this mean that the authors used a grid of EGM heights above WGS84?

*We rewrote this part for clarity as "The DEMs were converted to orthometric heights by subtracting the EGM2008 geoid and projected in UTM using the World Geodetic System (WGS1984) using zone 19S for Chile, zone 18S for southern/central Peru, and zone 17S for northern Peru/Ecuador.".*

Legend of Figure 3: There was a switch between x-axis and y-axis.

*We changed the caption of this figure (now Fig. 4).*

QR. Indicate that it varies between 1 and 5; there is an error on the 3rd coefficient: 0.4 rather than 0.4 * 1.2.

*The factor of 1.2 in the $3^{rd}$ coefficient is added to maintain the possibility of QR = 5. We added an explanation in lines 396-397.*

*We rearranged this part to make it more coherent with the QR equation, its range (1 to 5) is now indicated directly before (lines 375-376).*

Line 555. Is there a reason why error and dispersion are correlated (not clear to me)?

*We do not mention a correlation. The higher number of measurements results in a more accurate representation of the measurement errors, since less measurements sample a smaller part of the topography. We extended the explanation in line 631.*

Line 617. The authors must quote Macharé and Ortlieb, a key paper.

*We included this reference.*

Line 651-652. A ref to support this assertion is missing.

*We added the reference Anderson et al. (1999) and Trenhaile (2002).*

Line 667-671. It is possible that wave power is not the main driver for coastal erosion...

Since we observe a positive correlation between wave height/tidal range and the number of terrace measurements (Fig. 11), we infer that wave energy predominantly controls coastal erosion and marine terrace formation rather than rock resistance or other characteristics – this is in line with most geomorphic concepts on this subject.

**Additional References**

Anderson, R.S., Densmore, A.L., Ellis, M.A., 1999. The generation and degradation of marine terraces. Basin Research 11(1), 7–19. doi:10.1046/j.1365-2117.1999.00085.x.

Le Roux, J.P., Tavares Correa, C., Alayza, F., 2000. Sedimentology of the Rímac-Chillón alluvial fan at Lima, Peru, as related to Plio-Pleistocene sea-level changes, glacial cycles and tectonics. Journal of South American Earth Sciences 13(6), 499–510. doi:10.1016/S0895-9811(00)00044-4.

Lorscheid, T., Rovere, A., 2019. The indicative meaning calculator – quantification of paleo sea-level relationships by using global wave and tide datasets. Open Geospatial Data, Software and Standards 4(1), 591. doi:10.1186/s40965-019-0069-8.

Macharé, J., Ortlieb, L., 1992. Plio-Quaternary vertical motions and the subduction of the Nazca Ridge, central coast of Peru. Tectonophysics 205(1-3), 97–108. doi:10.1016/0040-1951(92)90420-B.

Pedoja, K., Husson, L., Johnson, M.E., Melnick, D., Witt, C., Pochat, S., Nexer, M., Delcaillau, B., Pinegina, T., Poprawski, Y., Authemayou, C., Elliot, M., Regard, V., Garestier, F., 2014. Coastal staircase sequences reflecting sea-level oscillations and tectonic uplift during the Quaternary and Neogene. Earth-Science Reviews 132, 13–38. doi:10.1016/j.earscirev.2014.01.007.

Pedoja, K., Husson, L., Regard, V., Cobbold, P.R., Ostanciaux, E., Johnson, M.E., Kershaw, S., Saillard, M., Martinod, J., Furgerot, L., Weill, P., Delcaillau, B., 2011. Relative sea-level fall since the last interglacial stage: Are coasts uplifting worldwide? Earth-Science Reviews 108(1-2), 1–15. doi:10.1016/j.earscirev.2011.05.002.

Regard, V., Saillard, M., Martinod, J., Audin, L., Carretier, S., Pedoja, K., Riquelme, R., Paredes, P., Hérail, G., 2010. Renewed uplift of the Central Andes Forearc revealed by coastal evolution during the Quaternary. Earth and Planetary Science Letters 297(1-2), 199–210. doi:10.1016/j.epsl.2010.06.020.

Trenhaile, A.S., 2002. Modeling the development of marine terraces on tectonically mobile rock coasts. Marine Geology 185(3-4), 341–361. doi:10.1016/S0025-3227(02)00187-1.

---

## Author Comment (AC2) · 12 Mar 2021

**Comments by Reviewer #2: Paula Marques Figueiredo**

However, there is room for improvement and I strongly encourage the authors to submit a revised final manuscript. I present a list of comments and minor corrections/ suggestions, and there are two main aspects that I would like to see properly addressed, especially in a manuscript dedicated to the last interglacial.

The authors only provide the example of Chala Bay (southern Peru) to illustrate their methodology. Chala Bay is probably a good example, since it exhibits clear morphology and has geochronology information.

> We added another example at Punta Galera to illustrate our methodology.

The authors identified the two terraces as MIS 5e and MIS 5c, and based their interpretation in Saillard (2008) data. Taking in consideration the inner edges elevations for the two terraces, and assuming them as MIS 5e and MIS 5c, the calculation of the uplift rates for each is different, implying uplift rates changes (using Sidall et al., 2007 and Creveling et al., 2017 for example). However, if we consider that the lower terrace is not MIS 5c but instead MIS 5a, then the uplift rate for both marine terraces is very similar, which is a plausible configuration. In fact, when looking at Figure 2A, it is possible to note that between the paleo-shorelines inferred for the two terraces illustrated, a third paleo-shoreline is easily visible between the swath profile in 74°16W and 74°17W and in close to the profile immediately south from the larger drainage in the Bay area. This other possible inner edge is thus located between the two paleo-shorelines presented. This immediately suggests that the less preserved morphology between the terraces can correspond to the MIS 5c and that the inner-edges at ~70±2 m and ~37±2 may instead be respectively MIS 5e and MIS 5a. A MIS 5c evidence, is only present when there were conditions to promote its preservation, since in most cases worldwide was indeed re-occupied by MIS 5a. Testing if possible less preserved feature is evidence of MIS 5c, can be performed for example, by estimating uplift rates and look for consistency with MIS 5e and 5a uplift rates. This is a relevant question for most MIS 5 users and needs to be clarified.

> There are several arguments against the presence of MIS-5a in this area. In the San Juan the Marcona area (north of Chala Bay) uplift rates are much higher compared to Chala Bay (~0.85 m/ka) and MIS-5a and 5c are apparently preserved, MIS-5a occurring at ~40 m altitude. Despite that this elevation is similar to the lower terrace level in Chala Bay as indicated by the reviewer, its presence would imply a localized increase in uplift rate between the substages MIS-5e and MIS-5a only in Chala Bay. Based on the shoreline-angle measurements from the ten profiles in Chala Bay, we obtain median uplift rates of 0.55 m/ka for MIS-5e and MIS-5c. This coincides with our assumption of steady uplift rates during the entire MIS-5. If we assign the lower terrace level to MIS-5a, we obtain a higher median uplift rate of 0.72 m/ka. We argue that pronounced changes in uplift rate in the limited time frame of MIS-5 are highly unlikely. On the other hand, the subtle features indicated by the reviewer at Chala Bay were previously described by Goy et al. (1992) as alluvial fans. We therefore do not agree with the parallel interpretation proposed by the reviewer. However, we do agree that clarification on this topic is important and included a paragraph for further information on the interpretation of the substages (lines 302-312).

Saillard (2008) in fact suggests the same, that the morphology could be a mix of MIS 5c and/or MIS 5a and ultimately favoring a MIS 5a interpretation: - "Dans la zone de ChalaâˇA ˇRTanakaâˇA Rˇ Chaviña, la terrasse marine à +60 m, dans la baie de Chala, a un âge supérieur à 100 ka, et correspond vraisemblablement au SI 5 et au SSI 5c ou a (Figure 84). La terrasse marine à +94 m, dans la baie de Chala, a un âge minimum de 220 ka et correspond au SI 7 et au SSI 7c. De même, la terrasse marine à +154 m dans la baie de Chala, a un âge minimum de 250 ka et correspondrait donc au SI 9 et au SSI 9a. La terrasse d'abrasion marine à +90 m, au niveau du Cap de Tanaka, a un âge de 171 ka _ 21 ka et

correspond au SI 7 et au SSI 7a (page 202, Saillard 2008)" - "Nous avons corrélé la terrasse +60 m au SSI 5e (122 _ 7 ka), conformément à Goy et al. (1992) et à nos âges 10Be obtenus sur un cône alluvial déposé sur la terrasse, indiquant un âge supérieur à 100 ka. En effet, la terrasse +60 m est le premier niveau étendu et séparé de la terrasse supérieure par un grand escarpement qui mesure _30 m. … A partir des corrélations de Goy et al. (1992), nous avons corrélé les terrasses marines inférieures +2 m, +11 m et +31 m, à l'Holocène moyen (SI 1 ; 6000 _ 1000 ans), au SSI 3 et au SSI 5a (Table 8)." (page 208, Saillard, 2008).

Note that the elevations of Saillard are not the same presented in this manuscript, which I think is a consequence of presenting TerraceM data (and maybe orthometric elevation), and this difference should in fact be checked. The MIS 5c or MIS 5a is a most relevant question. Why have the authors chosen to consider the morphology as MIS 5c? Is there any other evidence that supports it? As geochronology data? This also raises questions about the other locations where the authors also interpret MIS 5c landforms. If MIS 5c was recognized (and generally less preserved worldwide) why was it not possible to identify MIS 5a? If that was the case, what mechanisms differ from 112ka to 80 ka? And why is the Quality Rating for this location rated as high confidence?

Our assignment of mapped terrace levels to MIS-5c is primarily based on age constraints by Saillard et al. (2011) for the Marcona area and Jara-Muñoz et al. (2015) for the area between 34° and 38°S. However, in order to evaluate the possibility that our correlation with MIS-5c is flawed, we estimated uplift rates for the lower terraces by assigning them tentatively to either MIS-5a or MIS-5c. We interpolated the uplift rates derived from the MIS-5e level at the sites of the lower terraces and compared the differences (Figure R1A). If we infer that uplift rates were constant in time at each site throughout the three MIS-5 substages, the comparison suggests these lower terrace levels correspond to MIS-5c because of the smaller difference in uplift rate, rather than to MIS-5a (Figure R1B).

We added this information and the associated figure to the text to better illustrate this point and to address the reviewer's concern (lines 302-312, Figure 3).

[Figure]

Figure R1. Comparison of MIS-5 uplift-rate estimates. (A) Uplift rates derived by correlating mapped terrace occurrences located immediately below the MIS-5e level to either MIS-5c (blue) or MIS-5a (red) with respect to MIS-5e uplift rates. Marine terraces correlated to MIS-5c by an age constraint are plotted in gray color. (B) Histograms of differences between MIS-5a or MIS-5c uplift rates and MIS-5e uplift rates. Vertical lines show median uplift-rate differences.

> Terrace levels of MIS-5e, c, and a were only observed at San Juan de Marcona – an area with one of the highest uplift rate (~0.85 m/ka). We interpret that in areas below such high uplift rates the marine terrace morphology is dominated by MIS-5c rather than MIS-5a, probably not completely reoccupying MIS-5c. Instead, in areas with similar or higher uplift rates, the terrace morphology develops two separate terrace levels (MIS-5c and 5a).

Difficulties in characterizing MIS 5 and younger landforms bring me to the second major question. GIA causes a known impact to crustal vertical deformation that needs to be taken into account to infer paleo sea level positioning and decode tectonic imprint (e.g. Creveling et al., 2015, Simms et al., 2016). Comments regarding possible Glacio- Isostatic-Adjustments effects in the elevation of last interglacial and paleo sea-levels are absent from this manuscript. It is likely that theses effects along the Western South America coast may vary from null or very little to a certain amount, however, this has not been introduced and discussed and the reader is not aware if it was considered or excluded and if the annexed dataset took those effects in consideration.

> The amplitude and wavelength of GIA is mostly determined by the structure of the continental mantle and the crustal loads (Turcotte and Schubert, 1982). Therefore, GIA should not dramatically influence vertical deformation along non-glaciated coastal regions located in forearcs of active subduction zones. In addition, because of their intrinsic modelling complexities, we did not account for the GIA effect on terrace elevations and uplift rates.

> We added this to the manuscript in lines 422-427.

**List of comments:**

Line 38 – is there any difference between abrasion platforms and marine terraces? If so, can you explain what abrasion platforms are? Are marine terraces depositional landforms and abrasion platforms erosional? Line 60 – again what is the difference between marine terrace sequences and abrasion platforms? Clarifying may be useful for the reader. Maybe add definition of rasas, as well. Clarify what are the chosen nomenclatures for this manuscript.

> We removed the term abrasion platforms because this is used to define the modern landform, instead we use paleo-platform, which is an erosional feature.

> We introduced rasa surfaces in the marine terrace section (2.2.2.) in lines 194-196.

Lines 63-65 – "The marine terrace morphology comprises a gently inclined marine abrasion platform or depositional surface that terminates landward at a steeply sloping paleo-cliff surface. The intersection point between both surfaces represents the sea-level position during the formation of the marine terrace also known as shoreline angle" Please consider that it represents the higher sea-level associated to this landform and that it there may be a discrepancy between the timing of the shoreline angle trimming and the length of the MIS

> Considering that we are correlating the highest reach of the sea level during MIS with the shoreline angle, the temporal uncertainties will derive from the sea-level curve and how precise the sea-level highstand was determined. We only consider the duration of the highstand for the calculation of uplift rates, which is included in equation 4 (Gallen et al., 2014) (line 418).

Line 65 – please introduce the terminology Inner Edge, since it will be used in the later part of the manuscript and never explained.

> We added a definition in the methods section, where this term is used (lines 293-294).

Line 112 – "Several bathymetric anomalies have been recognized on the subducting Nazca plate." The bathymetric anomalies are landforms at the bathymetric surface, rather than on the plate. Please rephrase, explaining the landform, and if the landforms are in the continental shelf, abyssal plain? Why are they anomalies? Are they reliefs standing it out from a background? Are they seamounts or depressions, since both cases can be bathymetric anomalies? Since you will debate later that these features when subducted may cause an impact, consider describing their sizes, how high and wide are they (elevation from the ocean floor). Maybe include lithologies if you know (are they volcanoes, structural reliefs?)

> The described bathymetric features are geomorphic phenomena that have a higher topography with respect to their surrounding areas defined by the generally smooth bathymetric ocean-bottom surface; these features are therefore characterized as anomalies. In case of the subducting Nazca and Carnegie ridges they are seamounts resulting from hot-spot volcanism. We added this information and their dimensions to the text.

112-113 – "most prominent anomalies being subducted beneath South America are the Carnegie and Nazca" the anomalies are not subducted per se, but rather the morphologies"
Line 114 – "The Carnegie Ridge subducts roughly parallel with the convergence direction and its position should have remained relatively stable beneath the continent" please rephrase the word position and continent. The position is relative and it changes through time, consider using geometry, shape and for continent, consider renaming it with the concept of plate.
Line 119-120 "these bathymetric anomalies are thought to influence the characteristics of interplate coupling and seismic rupture" please consider rephrasing to something like these larger reliefs (seamounts etc? explain what they are) previously present at the surface of the now subducted plates are thought to ….

> We rewrote this part following your suggestions.

Line 124 and 126 – "amount of sediments" please consider rephrasing to volume of sediments, since amount could refer of number of different type of sediments, number of clasts… etc.
Line 127-128 –"thick trench sediments" consider rephrasing to thick trench sediment sequences. Is not sufficiently clear.
Line 141 – consider replacing dominates with is dominant or is prevalent.

> We prefer using "dominates".

Line 152 – "Smaller coastal fault systems" how small? You gave one order of magnitude for one example (2000 km) and now the reader doesn't know if small can be 800, 400, 50 km…

> We removed the term "small".

Line 139-157 – These paragraphs describe a lot of places that are not mentioned in figure 1 and the reader doesn't really know their location. In addition, although not the scope per se of this manuscript, consider adding a sentence of two, stating that the different kinematics reflect the deformation (different areas/ segments) associated to the subduction. Also, can you add reference to a figure? For someone who is unfamiliar with the Geography, this is hard to follow.

> We followed the suggestions by the reviewer adding more figure references and another sentence about kinematics reflecting the deformation associated with the subduction.

> The locations are indicated in Fig. 1 – in form of numbers described in the caption. In most cases the latitude is also mentioned in the text to further describe these locations.

Line 169-185 – These paragraphs describe a lot of places that are not mentioned in figure 1 and the reader doesn't really know their location, as for example Gulf of Guayaquil (3°S) and the Dolores-Guayaquil megashear, San Juan de Marcona area.

We added the Gulf of Guayaquil to the caption of figure 1. Please notice that the other locations were already included in the caption.

Based on comments for lines 139-157 and 169-185, would it be possible to add another map to Fig 1 with more information regarding sites and in land structures kinematics (suggestion, different colors for different kinematics)? It is very hard to read the small numbers referring to the location names.

Instead of making a new figure, we increased the size of the location numbers in figure 1.

Line 188 – what is rasa in comparison with abrasion platforms? Clarifying the different nomenclatures is very positive.

As explained earlier, we changed the term abrasion platform.

Line 222 – can you define what the Maule segment is? This nomenclature is not included in the map, and it has not been introduced earlier.

The Maule segment is the rupture zone of the 2010 Maule earthquake of M8.8. We changed this part to put more emphasis on the area and not on the segment name.

Line 228 – "extensive uplift rate of 0.31 to 0.42 m/ka", Can you add information for the period of time corresponding to this uplift rate, since MIS 5, since before? Add a time period.

The uplift rates refer to the time frame since MIS-5e (last interglacial). We added the time period.

Line 229 – 0.34 m/ka is not lower than 0.31 to 0.42 m/ka, is actually within the range. Can you rephrase, please?

We rephrased this sentence.

Line 230 – 0.17– 0.21 m/ka for what time period?

The uplift rates also refer to the time frame since MIS-5e (last interglacial). We added the time period.

Line 231-235 – "Marine Terraces above … last 125 ka according to.." this sentence is confusing, maybe rewrite to clarify. Consider adding the word between to help differentiate.

We rewrote this part.

Line 243 – "oldest Pleistocene shore platforms" can you provide information regarding the age?

We improved this part by adding more information on the age of the terraces and the timing of uplift-rate change. Lower to Middle Pleistocene shore platforms indicate a period of tectonic stability or subsidence followed by accelerated and spatially continuous uplift after ~400 ka (MIS-11).

Line 243- 246 – "The Central Andean rasa (15°–33°S) and the oldest Pleistocene shore platforms – which are also generally wider – indicate accelerated and spatially continuous uplift after a period of tectonic stability or subsidence. According to Melnick (2016), the Central Andean rasa has experienced

slow and steady long-term uplift at 0.13±0.04 m/ka during the Quaternary," It is my understanding from what is written that the Central Andean rasa were subjected to a low uplift rate, favoring their development. Is this long-term uplift at 0.13±0.04 m/ka still the same? Was the period of tectonic stability or subsidence prior to the Central Andean rasa? Or the uplift initiated already during the Quaternary, and if yes, when? This sentence can be improved for clarification. Remember that today and yesterday are Quaternary as well.

> For clarity, we rewrote parts of this paragraph. Our intention was to explain the two current hypotheses for coastal uplift rates. (1) Renewal of uplift rates since ~400 ka after a period of tectonic stability or subsidence (Regard et al., 2010; Rodríguez et al., 2013; Martinod et al., 2016) and (2) Slow and steady long-term uplift at 0.13±0.04 m/ka during the Quaternary (Melnick, 2016).

Comment to the entire section 2.2.2 – Would greatly benefit if supported by an additional figure with geographic and some structural as suggested earlier. The reader is not aware of what is the Maule segment for example. It is very hard to follow locations.

> Instead of making a new figure, we improved figure 1. We marked all locations and the most important fault systems in figure 1. We specified the extent of the Maule segment in the text and put more emphasis on the latitudinal range.

Line 299 – Please verify if the provided link is correct

> Thanks for checking the link. We corrected it.

Line 299 – "placed swath profiles of variable width perpendicular to the previously mapped inner edge" It has not been given sufficient explanation regarding the selection of sites to study and where to place the swath profiles. How many for region, etc.. I understand their spacing being also controlled by drainage density, but maybe a sentence here to add further information is helpful.

> We tried to place swath profiles all along the South American margin where marine terraces of MIS-5 occur. Their location criteria are also based on the drainage spacing. We added more information on the placement swath profiles in lines 321-323.

Line 304 – "show only minimal deviations of less than 0.5 m" is this deviation regarding vertical or horizontal? Please clarify.

> As explained earlier, we measured shoreline-angle elevations. Therefore, it is a vertical deviation. We rewrote part of this line for clarity.

Line 308 – "we measured 1843 and 110 shoreline-angle elevations" in a total of 1953? It seems that a word is missing after 1843. What is the difference between 1843 and 110?

> We measured 1843 shoreline-angle elevations of MIS-5e and 110 of MIS-5c. We changed this sentence accordingly.

Line 309 – "5e and MIS-5c terrace levels". How do you know that it is MIS 5c and not MIS 5a? As far as I understand, there is not a direct numerical dating for it. Please discuss this identification as MIS 5c.

> We replied to this comment earlier and added information to the manuscript in lines 302-312.

Figure 2 – This figure is extremely important, by being taken as an example of the methodology for the entire area. Due to its relevance, I have some comments. The shaded relief is not very visible, and the

color scheme is not necessary, also because you have information for the inner edges elevations along the swath profiles. Slope information is barely noticeable and probably not necessary.

> We modified this figure by adding another test site. We prefer keeping the style of the figure, since the topography and slope are crucial to identify the individual terrace levels representing the method used in this study.

It lacks relevant information, most needed to validate your MIS interpretations. One can observe several other terraces higher and older that the one with the inner-edge at ~70±2 m elevation. Those terraces are in fact the ones that were dated with Terrestrial Cosmogenic Nuclides by Saillard (2008) and providing a time constrain for the marine terrace interpretation. I was unable to find in the thesis a direct TCN numerical dating for MIS 5e in Chala Bay, but rather for MIS 7 and older, and I think the authors need to clarify this information in the manuscript. I strongly advise to add to this image the inner edges from previous terraces (as dashed lines for instances) and add locations of TCN samples with numerical ages (X±x ka).

> The aim of our study is to provide a comprehensive database of MIS-5 terrace elevations. We therefore consider the mapping of the other terrace levels to be out of the scope for the purpose of this study, because it would add more uncertainties and certainly the figure would be overloaded with information.

> Saillard (2008) determined a [10]Be minimum age from an alluvial fan deposited on the terrace level that we correlate to MIS-5e:

> "Nous avons corrélé la terrasse +60 m au SSI 5e (122 ± 7 ka), conformément à Goy et al. (1992) et à nos âges [10]Be obtenus sur un cône alluvial déposé sur la terrasse, indiquant un âge supérieur à 100 ka."

> Further evidence for a confident assignment to MIS-5e is provided by the more precise age constraints for terrace levels older than MIS-7 (Saillard, 2008) and by the morphostratigraphic interpretation of Goy et al. (1992). This additional information increases the reliability of the age constraint significantly, which is also reflected in the confidence value of the Chala site. We improved this part in the manuscript (lines 386-388).

> We added the age constraints that were used for the Chala and Punta Galera sites to the figure and added another column for the age constraints to Table 1.

I also question the sudden change elevation of the lower inner edge in the western most and southern most profiles. Is it a steep section of the cliff? Why in one drives the inner edge up and in the other lowers it? Why was not possible to measure the upper terrace one? Since the confidence level given to these coastal landforms as MIS 5e and MIS 5c is 5 (high confidence), a proper justification supported in evidences is required.

> We consider the change in elevation to be within the range of error of both profiles. It was not possible to measure the shoreline-angle elevation in the southwesternmost profile, due to the very short paleo-platform.

> The confidence level is subjected to the reliability of the age constraint, which is part of the overall QR equation – together with for instance the RP distance. These values are high in the Chala Bay due to multiple age constraints for several terrace levels using a well-established dating method and short distances to the RP, which ultimately lead to a high QR of 5. In Chala Bay, we base our MIS-5e interpretation on the dating of Saillard (2008) and the MIS-5c interpretation on a comparison of uplift rates. This does not differ from other regions and was added to the manuscript as suggested in a previous comment.

Figure 3D – is n= 1953 referring to what? terrace-elevation measurements? What is the 1843 referred earlier?

It is referring to the total number of shoreline-angle measurements as stated in the caption. 1843 is the number of MIS-5e shoreline-angle measurements.

Line 342- 343 – "The following equation illustrates how we calculated the individual parameters and the overall quality rating:" Besides identifying variables, the authors do not provide any explanation for the equation per se, and it would be most adequate to do so. Can you please provide the basis for this equation? Maybe including referencesas well.

We developed this QR equation. We rewrote these lines to make this clearer in the text.

Line 397-398 – Only now is explained what 1843 and 110 are. Please consider explaining this earlier. (line 308).

We corrected this issue in line 331 based on your previous comment.

Figures 5 to 8 – should probably be organized together, similar to what was done with figure 4. It would be easier to compare the different regions and then it will be easier to compare that figure with figure 4. Consider including location of the ridges previously mentioned as anomalies.

We used figures 6-9 mainly for the description of the terrace elevations along the margin and to highlight the possible influences from crustal faults. Therefore, we want to stick with the placement of the figures in accordance with the descriptions in the text. For the comparison of the four areas to each other and their relation to bathymetric anomalies, we already provide figure 5 and 11.

Line 542 "Our statistical analysis of mapped shoreline-angle elevations" for the entire area? Does it make sense, when there are so many tide and wave height variations, not to mention changes imposed by tectonics?

We tried to provide a general statistical overview for the terrace elevations and their associated errors here. This enables us to describe the individual areas with respect to these median values as well as the results from different topographic data sets.

Line 551-556 – "We observe the lowest errors from the 30 m TanDEM-X, slightly higher errors from the 1-5 m LiDAR data, and the highest errors from the 12 m TanDEM-X. This observation is counterintuitive as we would expect lower errors for topographic data sets with higher resolution. The reason for these errors is probably related to the higher number of measurements using the 12 m TanDEM-X (1564) in comparison with the measurements using 30 m TanDEM-X (50), which result in a higher dispersion (Fig. 10B)." These sentences will greatly benefit from a better explanation or rewriting. It is unclear if the authors are stating that 30 m TAnDEM-X is a better topographic dataset for this type of analysis, or if their results of error analysis result from their own sampling.

Although we expected higher errors for DEMs with lower resolution (e.g., 30 m TanDEM-X), we see no clear relationship between the error distribution and DEM resolution. We state that this has do with the number of measurements being much higher, for example for a 12 m TanDEM-X (1564) compared to a 30 m TanDEM-X (50), which results in a higher dispersion and a more accurate representation of the errors for the higher number of measurements. We improved this explanation in the text.

Line 627-629 "This emphasizes also the importance of last interglacial marine terraces with respect to currently active faults, which might be compared in the future with shortterm deformation estimates

from GPS or the earthquake catalog." Short term estimates from GPS are really very short in time and may not reflect long term rates expressed by the faults, and active structures may even be seismic silent for Holocene times. Maybe focus in discuss their validity as a Late Pleistocene marker for crustal vertical deformation.

> The comparison between short-term deformation (GPS, seismicity) and long-term deformation (MIS-5 uplift rates) is beyond the scope of this database-oriented paper, and will be the focus of a forthcoming process-oriented study. Here, we wanted to generate a rigorously collected database that will form the foundation for future work on coastal and tectonic processes in different sectors along the convergent South American margin.

Line 651-652 – "An increase of wave height and tidal range may lead to enhanced erosion and morphologically well-expressed marine terraces, which is consequently reflected in a higher number of measurements" This statement probably needs further evidence.

> We added two references to support this statement here. This had been discussed in lines 717-722.

Line 664- 669 – has the lithology effect been considered? Some lithologies are easier to trim that others and other are more easily eroded. Has structural control (bedding, etc) taken into account in this analysis? Has the cliff erosion and in particular the Holocene retreat been compared? Obviously a higher cliff retreat will erode terraces which don't necessarily mean that they were never formed.

> Although we discussed the potential role of tectonics and climate on formation of marine terraces, this manuscript is mostly focused on presenting a MIS-5 database and not on the genesis of marine terraces. There are many papers that have focused on this issue and we refer to them in the text. Our database will certainly motive further studies on this topic.

Line 693-694 "The marine terrace elevations display a median value of 30.1 m for the MIS-5e level and a median uplift rate of 0.22 m/ka for MIS-5e and 5c." Why presenting median values in such a dynamic and variable tectonic setting along 5000 km? Is it relevant as a conclusion?

> We consider these values to be a good reference for the South American margin that will help to identify areas with anomalous values, which may be associated with local processes. We provide a short summary of such anomalous regions immediately afterwards.

**Additional References**

Gallen, S.F., Wegmann, K.W., Bohnenstiehl, D.R., Pazzaglia, F.J., Brandon, M.T., Fassoulas, C., 2014. Active simultaneous uplift and margin-normal extension in a forearc high, Crete, Greece. Earth and Planetary Science Letters 398, 11–24. doi:10.1016/j.epsl.2014.04.038.

Goy, J.L., Macharé, J., Ortlieb, L., Zazo, C., 1992. Quaternary shorelines in Southern Peru a record of global sea-level fluctuations and tectonic uplift in Chala Bay. Quaternary International 15-16, 99–112.

Jara-Muñoz, J., Melnick, D., Brill, D., Strecker, M.R., 2015. Segmentation of the 2010 Maule Chile earthquake rupture from a joint analysis of uplifted marine terraces and seismic-cycle deformation patterns. Quaternary Science Reviews 113, 171–192. doi:10.1016/j.quascirev.2015.01.005.

Martinod, J., Regard, V., Riquelme, R., Aguilar, G., Guillaume, B., Carretier, S., Cortés-Aranda, J., Leanni, L., Hérail, G., 2016. Pleistocene uplift, climate and morphological segmentation of the Northern Chile coasts (24°S–32°S): Insights from cosmogenic 10Be dating of paleoshorelines. Geomorphology 274, 78–91. doi:10.1016/j.geomorph.2016.09.010.

Melnick, D., 2016. Rise of the central Andean coast by earthquakes straddling the Moho. Nature Geoscience 9(5), 401–407. doi:10.1038/ngeo2683.

Regard, V., Saillard, M., Martinod, J., Audin, L., Carretier, S., Pedoja, K., Riquelme, R., Paredes, P., Hérail, G., 2010. Renewed uplift of the Central Andes Forearc revealed by coastal evolution during the Quaternary. Earth and Planetary Science Letters 297(1-2), 199–210. doi:10.1016/j.epsl.2010.06.020.

Rodríguez, M.P., Carretier, S., Charrier, R., Saillard, M., Regard, V., Hérail, G., Hall, S., Farber, D., Audin, L., 2013. Geochronology of pediments and marine terraces in north-central Chile and their implications for Quaternary uplift in the Western Andes. Geomorphology 180-181, 33–46. doi:10.1016/j.geomorph.2012.09.003.

Saillard, M., 2008. Dynamique du soulèvement côtier Pléistocène des Andes centrales Etude de l'évolution géomorphologique et datations (10Be) de séquences de terrasses marines (Sud Pérou - Nord Chili), Université Paul Sabatier, Toulouse.

Saillard, M., Hall, S.R., Audin, L., Farber, D.L., Regard, V., Hérail, G., 2011. Andean coastal uplift and active tectonics in southern Peru: 10Be surface exposure dating of differentially uplifted marine terrace sequences (San Juan de Marcona, ~15.4°S). Geomorphology 128(3-4), 178–190. doi:10.1016/j.geomorph.2011.01.004.

Turcotte, D.L., Schubert, G., 1982. Geodynamics: Applications of Continuum Physics to Geological Problema. John Wiley, New York (450 pp.).

---

## Author Response (AR2)

**Point-by-point response to the editor comments:**

1.) Lines 333 – explicitly state the indicative meaning used for the marine terraces. Maybe include an additional table and reference that table here (I know you give a value on lines 726 but it isn't clear how much this value varies across your study area).

> We calculated the indicative meaning for each marine terrace measurement. The associated values (upper and lower limits of modern analog, reference water level, and indicative range) can be found in the WALIS database, which we refer to in line 371. The variation of the reference water level along the WSAC is shown in Figure 11F.
>
> In line 372 we added that we calculated the indicative meaning "for each marine terrace measurement" and we also added a new Table 2 with the median values representing the indicative meaning for the four sectors that we also used for the results section.

**Table 2. Median values and standard deviations (2σ) that represent the indicative meaning along the WSAC. The four sectors were chosen based on their main geomorphic characteristics (see results section).**

|  | Upper limit of modern analog [m] | Lower limit of modern analog [m] | Reference water level [m] | Indicative range [m] |
|---|---|---|---|---|
| Ecuador and northern Peru | 2.89 ± 0.16 | -1.78 ± 0.47 | 0.54 ± 0.21 | 4.66 ± 0.65 |
| Central and southern Peru | 2.98 ± 0.31 | -3.05 ± 0.52 | -0.03 ± 0.11 | 6.06 ± 0.90 |
| Northern Chile | 3.01 ± 0.15 | -2.89 ± 0.30 | 0.06 ± 0.08 | 5.90 ± 0.51 |
| Central Chile | 3.21 ± 0.19 | -3.03 ± 0.38 | 0.07 ± 0.11 | 6.25 ± 0.60 |

2.) I think GIA is currently not adequately discussed (as suggested in the earlier review by Paula Marques Figueiredo). I appreciate the efforts and the addition of Figure 3 (which definitely strengthens your argument) but some questions still remain that likely have bearing on your findings (and particularly your uplift rate calculations. I agree with you that the tectonic signal likely dominants the differences in marine terrace elevations, but as written, the manuscript seems to discount GIA, prematurely. For example, GIA can result in 10's of m in differences in sea level elevations across a N-S transect (e.g. Potter et al., 2003; Creveling et al., 2017; Simms et al., 2016). Figure 3 assumes a MIS5e RSL that is constant across the area, which it likely was not. Furthermore, what elevations did you use for "sea levels" during MIS5e for Figure 3? GIA-induced differences are even greater for MIS5c. This influence would also apply to your later estimates of uplift (line 630) and errors (line 420). The variability is likely 5-10 m for MIS5e but could be as much as 20-30 m across the WSAC during MIS5c. This should be acknowledged as a limitation of your uplift estimates (section 5.a).

I think much of this could be dealt with by adding to Lines 422-427. Maybe look at the global model of Creveling et al. (2017) to estimate how much variability would be expected across this margin and mention that this is still less than the X m of differential elevations found along the margin – you could even include that in your error terms as uncertainty for the sea level estimates.

> We followed your suggestion by adding the information that GIA may cause local differences in sea level of up to 30 m (Simms et al., 2016; Creveling et al., 2017) [lines 211-212]. However, the results of Creveling et al. (2017) suggest differences in sea-level due to GIA along the WSAC north of 40°S range from -2 to +2 m, supporting our statement that "the amplitude and wavelength of GIA is mostly determined by the flexural rigidity of the lithosphere (Turcotte and Schubert, 1982) and should therefore not severely influence vertical deformation along non-glaciated coastal regions (Rabassa and Clapperton, 1990) that are located in the forearc of active subduction zones." [lines 429-431]

[Figure]

**Figure 2B of Creveling et al. (2017) showing the predicted GIA since MIS-5a (left). Histogram of GIA estimates along the WSAC (1°N-40°S).**

> We caution that current GIA models use an oversimplified lithospheric structure defined by horizontal layers of homogeneous rheology, which might be appropriate for cratons and ocean basins, but not necessarily for the forearc regions of subduction zones. Therefore, assessing the contribution of GIA to RSL during MIS-5 is beyond the scope of this study as it would require a dedicated modeling experiment.

3.) The third is relatively minor, could you clarify the difference between your "inner edge" and the commonly used "shoreline angle" a little earlier in section 3.1. Maybe the description on line 315 of the inner edge could be moved to line 293, when it is first introduced.

> We define the inner edge in lines 294-295 as the location at the foot of the paleo-cliff where significant changes in slope occur. A short version of this definition is used in lines 354-355 to explain the difference to the shoreline-angle elevation. We would like to point out that these definitions are described in great detail in the methodological papers on the TerraceM software (Jara-Muñoz et al., 2016; 2019). We cite both papers, which are easily accessible in the international literature and refrain from repeating this published material.